# Covariance Density Neural Networks

Om Roy[1,*],   Yashar Moshfeghi[1],   Keith Smith[1]

[1]Computer and Information Sciences, University of Strathclyde, Glasgow, Scotland, United Kingdom, G1 1XQ

`o.roy.2022@uni.strath.ac.uk`,

## Abstract

Graph neural networks have re-defined how we model and predict on network data but there lacks a consensus on choosing the correct underlying graph structure on which to model signals. CoVariance Neural Networks (VNN) address this issue by using the sample covariance matrix as a Graph Shift Operator (GSO). Here, we improve on the performance of VNNs by constructing a Density Matrix where we consider the sample Covariance matrix as a quasi-Hamiltonian of the system in the space of random variables. Crucially, using this density matrix as the GSO allows components of the data to be extracted at different scales, allowing enhanced discriminability and performance. We show that this approach allows explicit control of the stability-discriminability trade-off of the network, provides enhanced robustness to noise compared to VNNs, and outperforms them in useful real-life applications where the underlying covariance matrix is informative. In particular, we show that our model can achieve strong performance in subject-independent Brain Computer Interface EEG motor imagery classification, outperforming EEGnet while being faster. This shows how covariance density neural networks provide a basis for the notoriously difficult task of transferability of BCIs when evaluated on unseen individuals, while providing a principled, tuneable control over the stability–discriminability trade-off via the inverse temperature parameter $\beta$.

## 1 Introduction

Graph Neural Networks (GNN) have served as an essential platform for the modelling of Network data (Gama et al., 2020; Scarselli et al., 2009; Veličković et al., 2018; Ruiz et al., 2021; Defferrard et al., 2016; Isufi et al., 2020; Kipf & Welling, 2017; Vignac et al., 2020). Much of the recent developments in GNNs have stemmed from Graph Signal Processing (GSP)(Ortega et al., 2018; Sandryhaila & Moura, 2013) and other areas of Network Science (Peixoto, 2015; Guimerà & Sales-Pardo, 2009; Domenico et al., 2013; Lambiotte et al., 2014; Arenas et al., 2006). A key component of GSP is the Graph Shift Operator (GSO), whose eigen-decomposition underlies spectral filtering operations analogous to the Discrete Fourier Transform.

However, in many application domains, including neuroscience, finance, and genomics, we observe multivariate signals *without* a known underlying graph topology. Standard GNNs require a pre-specified graph, and learning one from data can introduce instability, overfitting, and noise sensitivity, especially under limited samples. A fundamental limitation of GSP is that the GSO is typically unrelated to the graph signal data itself (Smith et al., 2019).

CoVariance Neural Networks (VNNs) (Sihag et al., 2022) address this by using the sample covariance matrix, computed directly from the data, as the GSO. This approach has intuitive links to Principal Component Analysis (PCA) and inherits desirable transferability properties of GNNs to data of different dimensions (Sihag et al., 2022). However, VNNs face two key limitations:

1. **No control over the stability–discriminability trade-off**: VNNs inherently discriminate signal differences lying in the *low-variance* eigenspace, while high-variance principal components remain indistinguishable. There is no mechanism to tune which spectral components the network can resolve.

2. **Noise sensitivity**: Since VNNs discriminate in the low-variance subspace, noise concentrated in these directions common in low signal-to-noise ratio (SNR) settings (such as EEG data) directly degrades performance.

In this work, we propose **Covariance Density Neural Networks (CDNNs)**, which resolve both limitations. We construct a density matrix $\rho(\mathbf{C}) = e^{-\beta\mathbf{C}}/\mathrm{Tr}(e^{-\beta\mathbf{C}})$ from the sample covariance matrix $\mathbf{C}$, treating $\mathbf{C}$ as a quasi-Hamiltonian in the space of random variables. Using this density matrix as the GSO introduces qualitatively new capabilities.

Density matrices have long been used in Quantum Mechanics and information theory (Borst & Theunissen, 1999; Wilde, 2013) to describe physical systems as Quantum States (Domenico & Biamonte, 2016; Acín et al., 2007; Braunstein & Caves, 1994; Zdeborová & Krzakala, 2016; Hübner, 1992). A density matrix is a positive semi-definite, self-adjoint matrix with a trace of one acting on the Hilbert Space of the system (Acín et al., 2007). This gives a probabilistic interpretation to the eigenvalues of the matrix where the eigenmodes represent different probabilistic states while also allowing an entropic interpretation. The Hamiltonian of the system determines the evolution of the system defined by a density matrix. Naturally, the graph Laplacian has appeared as a candidate to represent this Hamiltonian operator. However, the question remains, what if the true underlying graph structure is unknown?

In this vein, we show that the sample covariance matrix, under reasonable assumptions, can act as spectral surrogate to the Graph Laplacian and thus play the role of a *Quasi*-Hamiltonian. We propose a novel density matrix constructed from the sample covariance matrix and define convolutions on this operator as the basis of our Covariance Density Neural Networks (CDNN). As a consequence of this formulation we present novel information theoretic tools such as the multi-scale Von Neumann entropy for Covariance matrices, a measure of entropy that can be applied to singular, low-rank covariance matrices. We also create multi-scale filter banks on the sample covariance matrix which we show improves performance. Further, we show empirically and theoretically the importance of the $\beta$ (inverse temperature) parameter and how it allows us to control the discriminability and stability of the network.

Empirically, we show that CDNNs outperform VNNs in financial forecasting where the underlying covariance matrix may be informative (Wu et al., 2020). CDNNs also show enhanced performance in the analysis of neurological signals, in particular, we demonstrate strong performance in classifying unseen individuals' EEG brain signals in Motor Imagery (MI) tasks (Tangermann et al., 2012; Xu et al., 2022). This leverages the transferability Property of VNNs and combines it with the increased stability and discriminability of CDNNs to classify 4-class motor imagery signals when evaluation is done on a test individual not seen in training (Tangermann et al., 2012). We show that our approach outperforms benchmarks in the field of EEG signal classification such as EEGNet(Keutayeva et al., 2024) while being faster. This is a step towards real-time Brain Computer Interfaces (BCI) (Xu et al., 2022; Yang et al., 2021; Zhang & Liu, 2018; Zhang et al., 2019; 2023) where high accuracy on unseen individuals coupled with low training times are highly desirable.

## 2 Background and Motivation

**Overview**

**Graph Signal Processing in brief.** In classical signal processing, a time series is a sequence of values indexed by time, and the *shift operator* (delay by one sample) underpins the Fourier transform and filtering. Graph Signal Processing (GSP) generalises this idea: a *graph signal* assigns a value to each node of a graph, and the *Graph Shift Operator* (GSO), typically the adjacency or Laplacian matrix, plays the role of the shift. Multiplying a signal by the GSO "propagates" values along edges, and repeated application (powers of the GSO) defines graph convolution filters that mix information at increasing neighborhoods. The eigen-decomposition of the GSO provides a *graph Fourier basis* analogous to the DFT, enabling spectral analysis and filtering of signals on graphs.

**Graph Neural Networks as learnable graph filters.** Graph Neural Networks (GNNs) are essentially graph filters with learnable coefficients. A typical GNN layer can be written as $\sigma\left(\sum_{k=0}^{K} h_k \mathbf{S}^k \mathbf{x}\right)$, where $\mathbf{S}$ is the GSO, $h_k$ are trainable filter coefficients, and $\sigma(\cdot)$ is a point-wise non-linearity (e.g., ReLU). The coefficients

$\{h_k\}$ determine the spectral response of the filter (how strongly different graph frequencies are amplified or attenuated) and are learned from data via backpropagation. Thus, GNNs combine the representational power of polynomial graph filters with the flexibility of learned parameterization and non-linear activation functions.

**Covariance as a graph operator.** When no graph topology is available, the sample covariance matrix $\mathbf{C}_n$ offers a natural data-driven alternative: its $(i, j)$-th entry measures the linear association between variables $i$ and $j$, providing implicit "edge weights." Using $\mathbf{C}_n$ as the GSO means that filtering a signal propagates information according to the statistical dependencies in the data, and the eigenvectors of $\mathbf{C}_n$, which are the principal components, serve as the graph Fourier basis.

**Why a density matrix?** A standard covariance-based filter can only discriminate signal differences in the *low-variance* eigenspace (see Theorem 1). The density matrix transformation $\rho(\mathbf{C}) = e^{-\beta\mathbf{C}}/\mathrm{Tr}(e^{-\beta\mathbf{C}})$ *inverts* this: for $\beta > 0$, large covariance eigenvalues map to *small* density eigenvalues, shifting high-variance components into the discriminable region. The parameter $\beta$ thus provides explicit, tuneable control over which spectral components the network resolves and provides a noise-discriminability tradeoff. By combining multiple $\beta$ values in a filter bank, CDNNs can discriminate across *all* spectral scales simultaneously. The normalization by the trace induces scale invariance and stability to the operator while allowing interpretability and transfer of information-theoretic measures.

We now introduce formal definitions that link these frameworks together.

**Definition 1** (Graph Convolutional Filter). *Let $\boldsymbol{h} = [h_0, \ldots, h_K]^\top$ be filter coefficients. Let $\boldsymbol{S}$ be the GSO. A graph convolutional filter of order $K$ is the linear map*

$$\boldsymbol{H}(\boldsymbol{x}) \;=\; \sum_{k=0}^{K} h_k\, \boldsymbol{S}^k \boldsymbol{x} \;=\; H(\boldsymbol{S})\, \boldsymbol{x},$$

*where $\boldsymbol{H}(\boldsymbol{S}) = \sum_{k=0}^{K} h_k S^k$.*

**Definition 2** (Graph Fourier Transform (GFT) (Isufi et al., 2024)). *For a diagonalizable GSO $\boldsymbol{S} = \boldsymbol{V}\Lambda\boldsymbol{V}^{-1}$ with eigenvectors $V$ and eigenvalues $\Lambda$, the GFT of a graph signal $\boldsymbol{x}$ is $\tilde{\boldsymbol{x}} = \boldsymbol{V}^{-1}\boldsymbol{x}$, and the inverse GFT is $\boldsymbol{x} = \boldsymbol{V}\tilde{\boldsymbol{x}}$.*

**Definition 3** (Covariance and Sample Covariance). *Let $\boldsymbol{X} \in \mathbb{R}^d$ with mean $\mu$. The covariance matrix is given by*

$$\boldsymbol{C} = \mathbb{E}[(\boldsymbol{X} - \mu)(\boldsymbol{X} - \mu)^\top]. \tag{1}$$

*From $n$ samples $\boldsymbol{X}^{(1)}, \ldots, \boldsymbol{X}^{(n)}$, the sample covariance is defined as*

$$\boldsymbol{C}_n = \frac{1}{n} \sum_{k=1}^{n} \big(\boldsymbol{X}^{(k)} - \bar{\boldsymbol{X}}\big)\big(\boldsymbol{X}^{(k)} - \bar{\boldsymbol{X}}\big)^\top, \tag{2}$$

*where*

$$\bar{\boldsymbol{X}} = \frac{1}{n} \sum_{k=1}^{n} \boldsymbol{X}^{(k)}. \tag{3}$$

*As $n \to \infty$, the sample covariance converges to the true covariance: $\boldsymbol{C}_n \to \boldsymbol{C}$.*

**Definition 4** (coVariance Fourier Transform (VFT) (Sihag et al., 2022)). *Let $\hat{\boldsymbol{C}}_n = \boldsymbol{U}\Sigma\boldsymbol{U}^\top$ be the eigendecomposition of the sample covariance. Then the VFT of a random sample $\boldsymbol{x}$ is $\tilde{\boldsymbol{x}} = \boldsymbol{U}^\top\boldsymbol{x}$.*

Quantum mechanical density matrices establish a probability structure for system states (Rao, 1945; Domenico & Biamonte, 2016). They must be Hermitian ($\rho = \rho^\dagger$), positive semi-definite ($\rho \geq 0$), and have unit trace ($\mathrm{Tr}(\rho) = 1$).

Importantly, the following density matrix has recently been Proposed(Domenico & Biamonte, 2016):

$$\rho = \frac{e^{-\beta\mathbf{L}}}{Z}, \quad Z = \mathrm{Tr}(e^{-\beta\mathbf{L}}),$$

where $\mathbf{L}$ is the graph Laplacian, and $\beta > 0$ is a parameter controlling the diffusion.

This allows the calculation of the spectral entropy of these matrices where eigenmodes define probabilistic routes of diffusion and $\beta$ allows the calculation of the entropy at different scales. Importantly, the sub-additivity of entropy is also maintained, a feat not achieved previously (Araki & Lieb, 1970; Anand et al., 2011). Taking inspiration from this we observe that there is a distinct relationship between graph Laplacian's and inverse covariance (precision) matrices and that graph learning is linked to sparse covariance estimation (Dittrich & Matz, 2020).

It is established that the estimation of the combinatorial graph Laplacian (CGL) matrix from observed signals is usually performed via the CGL estimator (CGLE) (Pavez, 2022). Specifically, the estimated Laplacian matrix $\hat{\mathbf{L}}$ is obtained as

$$\hat{\mathbf{L}} = \arg\min_{\mathbf{L} \in \mathcal{L}_p(E)} \left( -\log\overset{\dagger}{\det}(\mathbf{L}) + \operatorname{tr}(\mathbf{L}\mathbf{C}_n) \right), \tag{4}$$

where $\mathcal{L}_p(E)$ denotes the set of valid combinatorial graph Laplacian matrices consistent with the edge set $E$, and $\mathbf{C}_n$ is the sample covariance matrix of the observed graph signals.

Note that the estimation problem in equation 4 is grounded in a tight spectral connection between the graph Laplacian and the covariance of graph–stationary signals. Under the Gaussian–Markov assumption $\mathbf{C}_n = \mathbf{L}^\dagger$, the covariance can be expressed as a spectral function of $\mathbf{L}$ and therefore *shares the same eigenvectors* (Dong et al., 2019; Pavez & Ortega, 2016). This means that, when the number of observations $n$ greatly exceeds the graph order $N$, the sample covariance $\mathbf{C}_n$ forms a consistent surrogate for the Laplacian eigenspace. The CGLE in equation 4 can thus be interpreted as selecting edge weights so that $\mathbf{L}$ aligns with the empirical second-order statistics while retaining the common eigenbasis. It is thus intuitive to interpret the sample covariance matrix as a *signed* or *dual* data-driven Laplacian. We refer the reader to Appendix A.6 for more details.

It has also been shown that the eigenvectors of the Laplacian optimally decorrelate signals in a Gaussian Markov random field (GMRF) model where the precision matrix (inverse covariance matrix) is defined as the Laplacian (Zhang & Florencio, 2013). This Property makes precision matrix eigenvectors valuable for tasks like signal compression. However, the covariance matrix $\mathbf{C}$ shares the same eigenvectors as the precision matrix (up to eigenvalue scaling), retaining the decorrelation Properties without requiring explicit inversion of the covariance matrix. Moreover, $\mathbf{C}$ directly encodes global pairwise dependencies and serves as a more practical graph shift operator in graph neural networks (GNNs) (Keriven & Peyré, 2019; Zügner & Günnemann, 2019; Isufi et al., 2024; Bruna et al., 2014), as it avoids the sparsity constraints of the precision matrix and the numerical instability associated with its estimation (lou, 2017; Mestre, 2008).

A real signal $x_i$ on a graph has Dirichlet energy

$$\mathbf{x}^\top \mathbf{L} \mathbf{x} = \sum_{(i,j) \in E} w_{ij}(x_i - x_j)^2 = \underbrace{\sum_i d_i x_i^2}_{\text{self}} + \underbrace{\left( -\sum_{(i,j) \in E} w_{ij}\, x_i x_j \right)}_{\text{interaction}},$$

where $d_i = \sum_j w_{ij}$ is the vertex degree. For centered random variables $x_i$ with covariance matrix $\mathbf{C}$,

$$\mathbf{x}^\top \mathbf{C} \mathbf{x} = \sum_i \operatorname{Var}(x_i)\, x_i^2 + \sum_{i \neq j} \operatorname{Cov}(x_i, x_j)\, x_i x_j,$$

the diagonal variances $\operatorname{Var}(x_i)$ supply node potentials while off-diagonal covariances provide pairwise couplings. Thus $\mathbf{C}$ may be viewed as a *quasi-Hamiltonian*; its spectrum governs the same collective modes that minimise Dirichlet energy for $\mathbf{L}$.

1

In certain use cases when the covariance matrix is used as a "graph" constructed from the data itself (i.e brain networks) the conceptualization of the covariance matrix as a "quasi-Hamiltonian" makes intuitive sense

---

[1]We emphasise that the term "quasi-Hamiltonian" is used as a spectral analogy—highlighting the role of $\mathbf{C}$ in shaping the system's spectral modes and governing signal diffusion, rather than implying a strict quantum-mechanical interpretation. The analogy is grounded in the shared mathematical structure: just as the Hamiltonian determines the Gibbs state in statistical mechanics, $\mathbf{C}$ determines our density operator.

and may have practical benefits (the Hamiltonian is usually considered the energy operator of the system in quantum mechanics). Furthermore, when there is no predefined underlying graph structure the covariance matrix can act as a *spectral surrogate* to the graph Laplacian. This is the motivation of Covariance Density Neural Networks. [2]

> **Takeaway**
>
> When no graph topology is available, we demonstrate that the sample covariance matrix can serve as a spectral proxy for the true graph Laplacian, effectively functioning as a **quasi-Hamiltonian** or *energy operator* in our density-matrix formulation.

## 3 Covariance Density Neural Networks

In this section we introduce the theoretical foundations of Covariance Density Neural Networks. We first introduce the Covariance Density Matrix and Filter.

**Definition 5** (Covariance Density Operator and Filter). *Define the density operator associated with a covariance matrix $C$ as:*

$$\rho(\boldsymbol{C}) = \frac{e^{-\beta \boldsymbol{C}}}{\text{Tr}(e^{-\beta \boldsymbol{C}})}.$$

*For an input vector $x$ and real-valued parameters $\{h_k\}_{k=0}^{K}$, the output of the Covariance Density filter is defined as:*

$$\boldsymbol{z} = \boldsymbol{H}(\rho(\boldsymbol{C}))x = \sum_{k=0}^{K} h_k \left[\rho(\boldsymbol{C})\right]^k \boldsymbol{x}.$$

Analogously, we can define the Covariance Density Perceptron and its filter bank representation as follows:

**Definition 6** (Covariance Density Perceptron). *Consider a Covariance Density filter $H(\rho(\boldsymbol{C}))$ and a given nonlinear activation function $\sigma(\cdot)$. For an input $\boldsymbol{x}$, the Covariance Density Perceptron is defined as:*

$$\Phi(\boldsymbol{x}; \boldsymbol{C}, \boldsymbol{H}) = \sigma(\boldsymbol{H}(\rho(\boldsymbol{C}))\boldsymbol{x}).$$

**Definition 7** (Multi-Scale Covariance Density Filter Bank/Layer). *For a covariance density filter bank consisting of $F_{out}$ scales with $F_{in}$ m-dimensional inputs for each $\beta_m$ the m-th scale is defined as:*

$$\boldsymbol{x}_{out}[m] = \sigma\left(\sum_{g=1}^{F_{in}} \boldsymbol{H}_g^{(m)}\big(\rho(\boldsymbol{C})\big)\,\boldsymbol{x}_{in}[g]\right),$$

*and the final multi-scale output of the layer is*

$$\boldsymbol{x}_{final} = A\big(\boldsymbol{x}_{out}[1],\,\boldsymbol{x}_{out}[2],\,\ldots,\,\boldsymbol{x}_{out}[F_{out}]\big),$$

*where $A$ is an aggregation operator (e.g., concatenation, summation, or mean).*

In order to show that graph neural networks are stable, the assumptions of integral Lipschitz continuity of the network's GSO frequency response is traditionally assumed. For Covariance Density Networks however, we explicitly derive the Composite Lipschitz constant with respect to the density transformation as it is instructive for explaining the role of the inverse temperature $\beta$ in the network.

---

[2]While they exist, graph learning or pruning approaches can introduce instability, over-fitting, and noise sensitivity, especially when the true connectivity is uncertain or the number of samples is limited. By contrast, using the sample covariance (and thus the density matrix) as a fixed surrogate (similar to the Laplacian) provides a stable, interpretable operator without the risk of learning spurious edges.

**Theorem 1** (Composite Lipschitz Conditions for covariance density Filters)**.** *The composite frequency response of a Covariance Density Filter is given by*

$$h(\rho(\lambda)) = \sum_{k=1}^{K} \frac{h_k e^{-\beta\lambda k}}{Z^k},$$

*where $h_k$ are finite coefficients, $\beta > 0$ is a tunable parameter, $Z_k = \sum_{i=1}^{m} e^{-\beta\lambda_i k}$ is the partition function, and $\{\lambda_i\}_{i=1}^{m}$ are the eigenvalues of the covariance matrix $C$. Assuming $\beta$ and $h_k$ are finite constants, the covariance density filter satisfies:*

*The Lipschitz condition:*

$$|h(\rho(\lambda_2)) - h(\rho(\lambda_1))| \leq \alpha|\lambda_2 - \lambda_1|$$

*The Integral Lipschitz condition:*

$$|h(\rho(\lambda_2)) - h(\rho(\lambda_1))| \leq \theta \frac{|\lambda_2 - \lambda_1|}{\frac{|\lambda_1 + \lambda_2|}{2}}, \quad \alpha \leq \frac{\theta}{\sup\left(\frac{|\lambda_1 + \lambda_2|}{2}\right)}$$

*For some $\theta > 0$ with $\alpha = \sum_{k=0}^{K} |h_k||\beta k|$.*

Figure 1: Composite Lipschitz Condition Empirical Validation

[3]

Importantly, the theorem holds for both positive and negative $\beta$. We assume composite Lipschitz bounds on the density-transformed eigenvalues, treating filter coefficients as fixed, even though learned filters may violate integral Lipschitz. Viewing $\rho(C)$ as a GSO, linear filters cannot remain stable and distinguish signals beyond their Lipschitz cutoff—i.e., high-variance principal components stay indistinguishable. Pointwise nonlinearities can help (Ruiz et al., 2020), but some high-frequency content still eludes discrimination (Pfrommer et al., 2021).

Observe, in the covariance density context, that the eigenvalues of the covariance matrix **C** determine high-variance vs low-variance directions in the eigenbasis. If we denote these eigenvalues by $\{\lambda_i\}_{i=1}^{N}$ in ascending order (i.e. $\lambda_1 \leq \cdots \leq \lambda_N$), then the density operator has eigenvalues $\{\rho_i\}$ that are decreasing functions of $\lambda_i$ when $\beta > 0$ and increasing when $\beta < 0$.

For the Covariance matrix itself **larger** $\lambda_i$ corresponds to *higher variance* directions. Consequently, for the covariance density filter $H(\rho(\mathbf{C}))$, the composite integral Lipschitz condition is equivalent to

$$\left| \rho_i \, \partial_{\rho_i}\big(h(\rho(\lambda_i))\big) \right| \leq |\beta| \sum_{k=0}^{K} |h_k||k| \sup\left(\frac{|\lambda_1 + \lambda_2|}{2}\right)$$

---

[3]Although it is rather un-intuitive to consider negative values of $\beta$, allowing its use allows a greater exploration of the spectral profile, nevertheless such interpretations are not missing in the literature, i.e consider the communicability matrix introduced by Estrada et al. (Estrada et al., 2012) or even the Softmax function (LeCun et al., 2015; Goodfellow et al., 2016).

this forces the derivative of the filter to shrink as $\rho_i$ grows. Consider the case when $\beta > 0$, larger $\lambda_i$ leads to smaller $\rho_i$, so one sees a role reversal, i.e. large $\lambda_i$ are mapped to small density-eigenvalue $\rho_i$. and small $\lambda_i$ are mapped to large density-eigenvalue $\rho_i$.

Hence, imposing an integral Lipschitz bound in the covariance density setting tends to flatten the filter for the large values of $\rho_i$—i.e. for the low-variance eigenvalues $\lambda_i$.

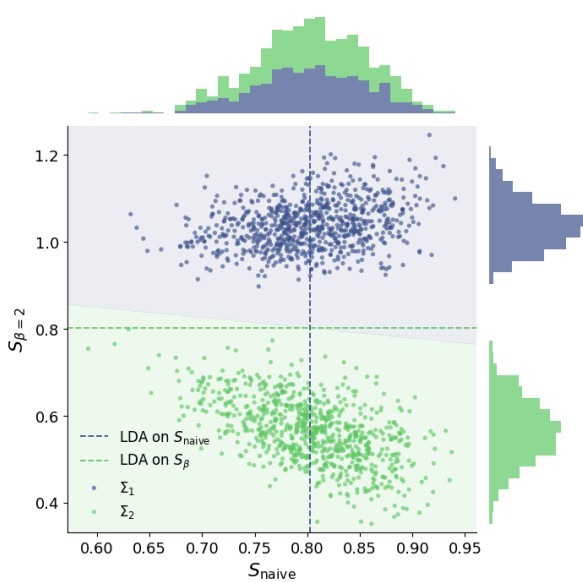

Figure 2: **Discrimination of two singular Gaussians using CVNE.** We sampled $N = 5 \times 10^4$ triples $(X, Y, Z)$, estimated covariances on sliding windows ($W = 128$) and computed $S_{\mathrm{naive}}$, $S_\beta$ ($\beta = 2$). Background: 2D LDA decision regions. Dashed: best 1D splits—Naive fails (AUC 0.5), von Neumann succeeds (AUC 1.0).

This leads to a different notion of "cut-off" frequency: instead of failing to distinguish signals whose difference lie in the highest-variance subspace, here the filter has trouble distinguishing components lying in the *lowest-variance* subspace. These non-discriminable subspaces are spanned by the eigenvectors whose variance in the covariance matrix is lower. In a PCA interpretation, the high-variance principal components become the "low-$\rho_i$ modes" and the covariance-density filter is able to discriminate signals that differ primarily in these high-variance directions.

The trade-off between stability and discriminability thus flips, precisely because $\lambda_i$ and $\rho_i$ are inversely related under the inverse exponential map.

For positive $\beta$ the exponential map ensures a 'compression' of high variance components into the discriminable region of the integral Lipschitz filter while for $\beta < 0$ we return to the original covariance setting where signal *differences* that lie in the low variance components eigenbasis can be discriminated but this time with a larger eigenvalue gap for higher variance components. i.e. the filter behaves similarly to a covariance filter. Importantly we can now construct multi-scale filter banks (and thus networks) that are discriminative in both high and low frequencies while being simultaneously more stable, although this increased discriminability could introduce noise into the system. The set of *discriminable* signals for a

CDNN is thus (Pfrommer et al., 2021):

$$
D_\Phi = \begin{cases} \big\{ (\mathbf{x}, \mathbf{y}) \in \mathbb{R}^N : \Phi(\mathbf{x}; \rho(\mathbf{C})) - \Phi(\mathbf{y}; \rho(\mathbf{C})) \in (V_{\text{low-}\lambda}) \big\}, & \beta > 0, \\[2mm] \big\{ (\mathbf{x}, \mathbf{y}) \in \mathbb{R}^N : \Phi(\mathbf{x};\ \mathbf{C}) - \Phi(\mathbf{y}; \mathbf{C}) \in (V_{\text{high-}\lambda}) \big\}, & \beta < 0. \end{cases}
$$

where $(V_{\mathcal{K}})$ corresponds to the eigenspace associated to those density-eigenvalues $\rho_i$ above the relevant cut-off. As these two spaces are orthogonal complements of each other, CDNNs constructed using multi-scale filter banks consisting of positive and negative $\beta$ values can discriminate signal differences in both high and low frequencies.

In this sense, it is also intuitive to allow the network to learn $\beta$ values from the data itself. In this vein, we show that there is no significant drop in performance when this parameter is learned from the data and not hand-picked using domain knowledge (Appendix A.3.1).

As shown in Theorem 1 the Lipschitz constant of the filter frequency response is explicitly controlled by $\beta$ and the filter coefficients. This is particularly useful as we can now directly manage the trade-off between the discriminability of the eigenvalues of the covariance matrix and the filters stability by tuning $\beta$.

Another benefit of the density matrix formulation is that it enables the use of information-theoretical methods.

Given that the eigenvalues of the density matrix now form a probability distribution (i.e. summing to 1) we can now define a multi-scale Von Neumann Entropy for Covariance matrices (CVNE).

This allows us to determine the 'regularity' of a covariance matrix. we show several desirable properties of this formulation, such as its ability to provide entropy values for low-rank, singular matrices, thus the CVNE and consequently CDNN's behave as inherent *regularizers*, stabilizing ill-conditioned covariance structures. We also prove the sub-additivity of CVNE (Appendix A.7) and show cases where it can be discriminative

> **Multiscale von Neumann Entropy for Covariance Matrices**
>
> Let $\mathbf{C} \in \mathbb{R}^{N \times N}$ be a positive semidefinite covariance matrix. The *Multiscale von Neumann Entropy* of $\mathbf{C}$ is
>
> $$S_\beta(\mathbf{C}) = -\operatorname{Tr}\big(\rho(\mathbf{C}) \log \rho(\mathbf{C})\big).$$

compared to naively normalizing the covariance matrix by its trace. For example, such an entropy is less effective when near global variance fluctuations occur (Appendix A.8). As the limiting extreme case of pure global scaling Figure 2 shows that it is in fact *scale-blind*.

For completeness Theorem 2 shows that Covariance Density Neural networks are permutation equivariant, another typical requirement of a suitable GNN.

**Theorem 2** (Permutation Equivariance of the Covariance Density Filter). *For any permutation matrix $T \in \mathbb{R}^{m \times m}$, define the permuted Covariance matrix $\hat{\boldsymbol{C}} = \boldsymbol{T}^T \boldsymbol{C} \boldsymbol{T}$ and the permuted signal $\hat{\boldsymbol{x}} = \boldsymbol{T}^T \boldsymbol{x}$. Then the filter $\boldsymbol{H}(\rho)$ is permutation equivariant, i.e.,*

$$\boldsymbol{H}((\rho(\hat{\boldsymbol{C}}))\hat{\boldsymbol{x}} = \boldsymbol{T}^T \boldsymbol{H}((\rho(\boldsymbol{C}))\boldsymbol{x}.$$

While the sensitivity of the ensemble covariance matrix to perturbations is well-studied we must derive new bounds in this new covariance density representation.

For this, we begin by relating the operator norm of the covariance matrix with the covariance density matrix.

**Lemma 1** (Error Bound for Covariance Density Matrix). *Let $\mathbf{C}$ be a covariance matrix and $\delta\mathbf{C}$ its perturbation. Further, let $\mathbf{E}$ be the density "error matrix", then*

$$\|\mathbf{E}\| \leq \frac{|\beta|\,\|\delta\mathbf{C}\|\,F(\beta, \mathbf{C}, \delta\mathbf{C})}{R}\Big(1 + m\,e^{\mathbf{1}_{\{\beta<0\}}|\beta|\,\|\mathbf{C}\|}\Big),$$

*where $R > 0$, $m = \dim(\mathbf{C})$, $\mathbf{1}_{\{\beta<0\}}$ is the indicator of $\beta < 0$, and*

$$F(\beta, \mathbf{C}, \delta\mathbf{C}) = \begin{cases} 1, & \beta \geq 0, \\[2mm] \dfrac{e^{|\beta|\|\mathbf{C}\|}\big(e^{|\beta|(\|\mathbf{C}+\delta\mathbf{C}\|-\|\mathbf{C}\|)} - 1\big)}{|\beta|\,(\|\mathbf{C} + \delta\mathbf{C}\| - \|\mathbf{C}\|)}, & \beta < 0, \end{cases}$$

*with $\lim_{\beta\to 0} F(\beta, \mathbf{C}, \delta\mathbf{C}) = 1$.*

**Theorem 3** (Stability of Covariance–Density Network). *Consider an $L$-layer, $F$-channel multi-scale CDNN $\Phi(\mathbf{x}; \widehat{\mathbf{P}}_N, \mathbf{H})$ with Lipschitz constants $\{\alpha_i\}$ and inverse-temperatures $\{\beta_i\}$. Let $Q > 0$ be a constant. Assume $|h_{fg}^{(\ell)}(\rho)| \leq 1$, $\sigma$ is 1-Lipschitz, and let $\kappa$ be as in (Sihag et al., 2022). For any $\varepsilon \in (0, \frac{1}{2}]$, with probability $\geq 1 - n^{-2\varepsilon} - 2\kappa m/n$,*

$$\big\|\Phi(\mathbf{x}; \widehat{\mathbf{P}}_N, \mathbf{H}) - \Phi(\mathbf{x}; \mathbf{P}, \mathbf{H})\big\| \leq L\Big(\prod_{i=1}^{F} \alpha_i . Q\big(\underbrace{\tfrac{\|\delta\mathbf{C}\|}{R}\overbrace{F(\beta_i, \mathbf{C}, \delta\mathbf{C})}^{\text{penalty for }\beta<0}}_{\text{density estimation error}}$$

$$\times \underbrace{\sqrt{m}\,\big(1 + m\,\overbrace{e^{\mathbf{1}_{\{\beta_i<0\}}|\beta_i|\|\mathbf{C}\|}}^{\text{penalty for }\beta<0}\big)}_{\text{density estimation error}} + \underbrace{\tfrac{m}{n^{\frac{1}{2}-\varepsilon}}}_{\text{dimensionality impact}}\big) + O\big(n^{-1}\big)\Big)^{L-1}.$$

From the stability bound, the terms $F(\beta, \mathbf{C}, \delta\mathbf{C})$ and $\exp\{\mathbf{1}_{\{\beta<0\}}|\beta|\|\mathbf{C}\|\}$ quantify penalties associated with negative $\beta$ in the uncertainty in the covariance density estimate, amplifying the covariance uncertainty. These

terms vanish for positive beta indicating the network is more stable for positive values of beta. As expected, the uncertainty of the covariance density estimate is controlled by the Lipschitz constant $\alpha$ which, recall, is controlled completely by the filter coefficients and beta. Thus as beta tends to 0 the network becomes increasingly stable. However each filter in the filter bank of size $F$ can have a different $\beta_i$ thus including larger or negative values of $\beta$ will reduce the stability of the network. Also note, that even if there are large errors in the estimation of the true covariance matrix (indicated by $\delta\mathbf{C}$) this can be offset by using smaller values of $\beta$, confirming the de-noising effect. The dimensionality impact term $\frac{m}{n^{\frac{1}{2}-\varepsilon}}$ reflects the relationship between the sample size $n$, the dimensionality $m$, and the convergence rate.

> **Takeaway**
>
> We propose multi-scale Covariance Density Neural Networks (CDNNs), show that $\beta$ controls the stability-discriminability tradeoff of the network, and formulate the Multiscale Von Neumann Entropy for Covariance Matrices.

## 4 Experimental Results

### 4.1 Covariance Matrix Construction

**Financial data.** Each asset/feature constitutes a random variable (node), and each time step within a sliding window constitutes one observation. The sample covariance matrix is computed once over the full training set and captures cross-asset correlations.

**EEG data.** Each EEG channel is a random variable (node), and each time step in a trial constitutes one observation. The covariance matrix is computed once over all training subjects' trials timesteps, repeated for all participants and averaged over, producing a global cross-channel covariance that captures shared spatial patterns across individuals.

### 4.2 Financial Forecasting

Covariance matrices have been deemed useful in predicting financial asset behaviour, offering insights into asset interactions and enabling better risk management strategies. Building on this, covariance-based neural networks show significant promise, this promise however, is largely affected by the low signal to noise ratio (SNR) present in financial data and thus the estimates of it's underlying covariance structure.

The S&P 500 dataset(Jander, 2023) spans 10 years of multivariate time series data, including features like stock volatility and daily option volumes. The Exchange Rate dataset (Wu et al., 2020) consists of daily exchange rates for different currencies across 7588 open days. Covering the period from January 2, 2020, to February 2, 2024, the US Stock dataset(Kumar, 2024) includes various variables across major stocks, indices, commodities, and cryptocurrencies, offering insights into pricing and trading activity.

Table 1 compares traditional graph-based models (e.g., Graph Attention Networks (GAT), Graph Isomorphism Network (GIN), Graph Convolutional Networks (GCN))) with covariance-based models (VNN, CDNN). We also include a hybrid model that first applies the covariance density filter at multiple scales before applying an attention layer.

We attribute the improved performance of CDNN over a standard VNN to the versatility of the multi-scale filter bank, capturing information at different scales. Over all datasets, either the CDNN or hybrid model performed the best. This indicates that the added covariance information is indeed informative and that enhancing attention based models with this information improves performance.

The robustness of covariance models likely stems from their ability to leverage stable cross-asset correlations driven by macroeconomic or sector factors. This inductive bias makes them inherently *risk-aware*, ensuring better diversification and consideration of correlated risks. In addition, the noise-robustness of CDNNs shows improved performance over all datasets.

Table 1: Mean Absolute Error (MAE) ± std dev over 3 seeds ↓ for horizons 1, 3, 5. **Best**; *underline second best*.

| Data | Mthd | H1 | H3 | H5 |
|---|---|---|---|---|
| Exchange | VNN | 0.1336±0.0075 | 0.1677±0.0212 | 0.1786±0.0070 |
| | CDNN | *0.1102±0.0033* | *0.1231±0.0065* | *0.1359±0.0137* |
| | GCN | 0.2415±0.0109 | 0.2334±0.0034 | 0.2371±0.0177 |
| | GIN | 0.1939±0.0089 | 0.1950±0.0031 | 0.2075±0.0151 |
| | GAT | 0.1819±0.0245 | 0.1852±0.0239 | 0.2075±0.0228 |
| | Hybrid | **0.1057±0.0073** | **0.1171±0.0120** | **0.1257±0.0077** |
| S&P 500 | VNN | 0.6198±0.0184 | 0.6392±0.0016 | 0.6553±0.0095 |
| | CDNN | 0.5848±0.0031 | **0.5983±0.0049** | **0.6090±0.0026** |
| | GCN | 0.6031±0.0193 | 0.6399±0.0007 | 0.6719±0.0094 |
| | GIN | 0.6682±0.0115 | 0.7314±0.0123 | 0.7103±0.0090 |
| | GAT | 0.5930±0.0245 | 0.6309±0.0151 | 0.6503±0.0198 |
| | Hybrid | **0.5772±0.0038** | *0.6000±0.0075* | *0.6111±0.0027* |
| US Stock | VNN | 0.4244±0.0159 | 0.4822±0.0083 | 0.6056±0.0063 |
| | CDNN | **0.3412±0.0179** | **0.4375±0.0129** | **0.5290±0.0185** |
| | GCN | 0.5572±0.0366 | 0.5934±0.0259 | 0.6592±0.0615 |
| | GIN | 0.6532±0.1248 | 0.6627±0.0955 | 0.7113±0.0907 |
| | GAT | 0.4346±0.0503 | 0.5141±0.0542 | 0.5832±0.0275 |
| | Hybrid | *0.3542±0.0058* | *0.4506±0.0115* | *0.5547±0.0211* |

### 4.3 Transferability of covariance density Networks in Brain Computer Interfaces

Brain-Computer Interfaces (BCIs) have long faced challenges in transferability (Zhang et al., 2019; Zhang & Liu, 2018; Yang et al., 2021), i.e., ensuring a model trained on multiple individuals generalizes to unseen individuals' brain signals. The sample covariance matrix computed purely over training individuals offers a promising solution by capturing stable, global correlations common across individuals. Using this as a shift operator for unseen individuals allows individual differences to be "filtered" by these shared patterns (Roy et al., 2024; 2023), this can be best related to the neuro-scientific concept known as *functional alignment* (Bazeille et al., 2021).

Our model is bench-marked on two BCI tasks: the 4-class BCI-2A motor imagery dataset (9 subjects) and the PhysioNet 2-class dataset (105 subjects). For BCI-2A, we train on data from all but one held-out subject, rotating through each as the test individual. On PhysioNet, we evaluate across 10 non-overlapping folds, ensuring each fold's subjects are unseen during training. In both cases, we compute a global covariance matrix from the training subjects and use it as a graph-shift operator to classify the held-out test data.

Table 2: Average Classification accuracies (%) ± std dev over subjects/folds and training speed (s/epoch) on BCI 2A (4 Class). **Best**; *Second best*.

| Method | 2A (2) | 2A (4) | PhysioNet | Average | Speed |
|---|---|---|---|---|---|
| GIN | 51.1 ± 3.33 | 32.6 ± 2.78 | 78.9 ± 4.03 | 54.20 | 0.91 |
| GAT | *62.67 ± 9.36* | 44.1 ± 6.77 | **81.0 ± 5.02** | *64.10* | 2.20 |
| VNN | 59.3 ± 4.86 | 30.1 ± 2.72 | 79.4 ± 3.01 | 56.30 | 0.90 |
| CDNN | **73.4 ± 8.53** | **50.4 ± 7.13** | *80.02 ± 2.74* | **67.90** | 1.10 |
| EEGNet | 61.4 ± 6.09 | *45.2 ± 5.47* | 75.1 ± 0.29 | 60.60 | 4.40 |
| CSP+LDA | 56.1 ± 4.21 | 33.37 ± 8.99 | 52.08 ± 3.86 | 47.20 | N/A |

Table 2 presents our results, comparing CDNNs with a VNN, more traditional Graph Based approaches such as GIN and GAT, a CSP approach and the lightweight EEGNet benchmark. For each graph model, we take the convolved signals—one time series per channel—and concatenate every channel's full sequence of time points into a single long feature vector in order to retain the high temporal resolution of EEG signals. That flattened vector is then fed into a simple fully-connected network (MLP), and we apply a Tanh nonlinearity followed by the classification layer.

The classical CSP + LDA approach performed the worst, EEGnet in fact underperformed on the PhysioNet dataset compared to simpler graph based models highlighting that, for *subject-independent* BCIs, less complex models are preferable to avoid over-fitting. While GAT tops the PhysioNet leader board, CDNNs are the strongest model overall, outdoing VNN both in accuracy and efficiency (1.1 s/epoch vs. EEGNet's 4.5 s), making them a suitable choice for rapid retraining and real-world BCI deployment.

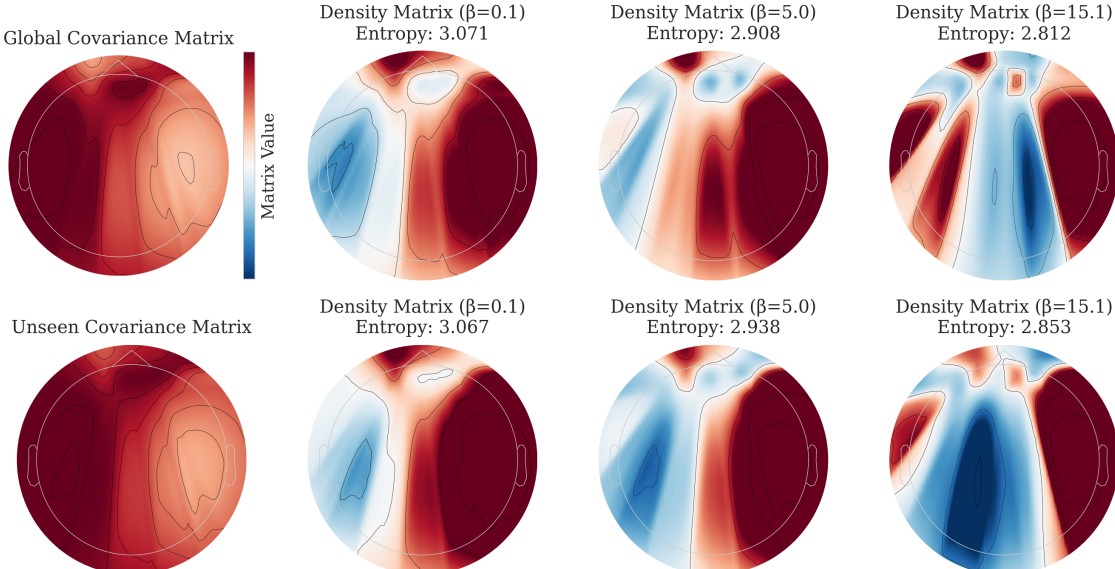

Figure 3: Contour Plot showing distribution of Covariance and Covariance Density Matrix on the Scalp at different scales. The top row indicates the averaged training covariance matrix (Excluding the Unseen Individual) and the bottom row indicates the unseen Individuals test covariance matrix. We can see that the training covariance matrix is relatively similar to the Unseen Individual and that different scales unveil further potential similarities thus increasing classification performance

The large gains on the BCI-2A 2-class and 4-class datasets are explained by the multi-scale filter bank's ability to capture patterns at different spectral scales of the cross-subject covariance matrix (Figure 3) while also being robust to noise at positive values of $\beta$ (Theorem 3).

### 4.4 Computational cost.

The primary additional cost of CDNNs over VNNs is the matrix exponential $\exp(-\beta\mathbf{C}_n)$. Since the covariance matrix is fixed, this is computed *once* via eigen-decomposition at $O(m^3)$ cost and cached. Subsequent forward and backward passes have identical per-step cost to VNNs: $O(Km^2)$ per order-$K$ filter, where $m$ is the number of channels/variables. Memory overhead is a single additional $m \times m$ matrix per $\beta$ scale. Empirically, with caching enabled, CDNN runtime matches VNN even for $m > 1000$ (see Appendix A.5, Figure 7). Compared to non-graph baselines, CDNN trains at $1.1\,\mathrm{s/epoch}$ on BCI-2A versus EEGNet's $4.4\,\mathrm{s/epoch}$—a $4\times$ speedup (Table 2).

## 5 Conclusions and Limitations

We introduce Covariance Density Neural Networks (CDNNs), which treat the covariance matrix as a Hamiltonian-like object to build a density matrix that acts as a graph shift operator in a Graph Neural Network. We showed that the sample Covariance Matrix can act as a spectral surrogate to the true graph Laplacian and thus constructed a novel density matrix, allowing us to use tools from information theory to define a new entropy measure for covariance matrices with notable benefits. We've also established rigorous stability bounds for the Covariance Density Filter and the network itself, showing how $\beta$ explicitly controls the stability of the network.

In real-world applications where covariance structures matter (e.g., financial or neurological networks), CDNNs show strong performance against traditional benchmarks and deliver significant gains over VNNs. In subject-independent motor imagery classification for Brain-Computer Interfaces (BCIs), they outperform EEGNet while being significantly faster. Although we relied on simple low-parameter architectures here, we believe that folding CDNNs into more complex neural network architectures can yield further benefits.

There are some clear limitations in the use of the Covariance matrix as it only encodes second-order statistics, it misses higher-order or non-linear dependencies. Future work could integrate mutual-information matrices or Gaussian-process kernels into the density matrix formulation. Furthermore, time-varying or non-stationary signals (e.g. drift in EEG) need dynamic covariance models or online adaptation and exploration beyond static matrices should be explored and can be particularly beneficial in the study of Dynamic Functional Connectivity (DFC) in the brain. We have to also consider that the computational cost scales with the size of the covariance matrix and matrix exponentials thus may become harder to compute. Since we only need the product of the matrix-vector multiplication approaches such as Krylov Subspace Approximation methods may be useful.

## 6 Impact Statement

This work may enhance Brain-Computer Interfaces (BCIs) by enabling more reliable brain signal decoding for assistive technologies. However, ensuring secure, anonymized handling of neurological data and addressing fairness across diverse users remain important challenges to prevent biases in medical and neuro-technological applications.

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

# A  Appendix

## A.1  Hardware Specifications

All experiments were performed on an NVIDIA A100 Tensor Core GPU. The A100 features 6 912 CUDA cores, 3 456 FP64 CUDA cores, and 432 third-generation Tensor Cores, delivering up to 19.5 TFLOPS of FP32 throughput. It is equipped with 40 GB of HBM2e memory and offers a peak memory bandwidth of approximately 1.6 TB/s

## A.2  Hyperparamter Selection and Model Details

Table 3: Hyperparameters for Financial Datasets

| Model | Learning Rate | Epochs | Batch Size | Hidden Dim | Num Layers | Activation | Dropout | Betas |
|---|---|---|---|---|---|---|---|---|
| GAT | 0.001 | 500 | 64 | 128 | 1 | ELU | 0.2 | N/A |
| VNN | 0.001 | 500 | 64 | 128 | 1 | ELU | 0.2 | N/A |
| CDNN | 0.001 | 500 | 64 | 128 | 1 | ELU | 0.2 | Learned 4 $\beta$ filterbank, init: [-0.01,0.01,0,0] |
| GIN | 0.001 | 500 | 64 | 128 | 1 | ELU | 0.2 | N/A |

4

Table 4: Hyperparameters for BCI 2A and PhysioNet Datasets

| Model | Learning Rate | Epochs | Batch Size | Hidden Dim | Num Layers | Activation Functions | Dropout | Betas/Order |
|---|---|---|---|---|---|---|---|---|
| EEGNet | 0.001 | 500 | as in (Lawhern et al., 2018) | as in (Lawhern et al., 2018) | as in (Lawhern et al., 2018) | as in (Lawhern et al., 2018) | as in (Lawhern et al., 2018) | N/A |
| VNN | 0.0001 | 50 | 64 | 128 | 1 | Tanh | 0.7 | N/A |
| CDNN | 0.0001 | 50 | 64 | 128 | 1 | Tanh | 0.7 | [0.1, 5.0, 15.1] |

## A.3  Data Availability

We used the TorchEEG Python package (Zhang et al., 2024) with the Mother of all BCI Benchmarks (MOABB)(Aristimunha et al., 2023) wrapper for the Motor Imagery data. The BCI 2A data is also available freely at https://www.bbci.de/competition/iv/. The Physionet dataset is available at https://physionet.org/about/database/. The exchange rate dataset is from (Wu et al., 2020) and was obtained from the Github repository for (Cavallo et al., 2024).The S&P 500 and US Stock market and commodities data are available freely from (Jander, 2023) and (Kumar, 2024). For the US Stock dataset only the price columns for each commodity was kept.

### PhysioNet Dataset

This dataset comprises EEG recordings from 109 healthy volunteers. In each trial, a visual target appears on either the left or right side of a display. The participant then mentally rehearses opening and closing the corresponding hand until the target disappears, and subsequently relaxes. Data were acquired using the BCI2000 system with 64 EEG channels, sampled at 160 Hz, and each trial has a duration of 3.1 s. Recordings from subjects 88, 89, 92, and 100 were excluded due to technical issues and excessive rest-state activity, resulting in a final cohort of 105 participants, each completing approximately 45 trials. We ran experiments on a 10 fold split of all individuals, where the test fold participants are unseen during training. All experiments were performed on the same randomization seed for all models for reproducible results.Raw EEG measurements without any preprocessing were used on this data set excpet what was already done in (Goldberger et al., 2000).

---

[4]All Graph Based models treats temporal data as node features on the underlying graph. For GAT and GIN the underlying graph a fully-connected network. For CDNN and VNN the underlying graph is the sample covariance matrix and is computed once over the complete training set and fixed during training.This matrix is then untouched and used for both training and testing. For VNN's the matrix is trace normalized for a fair comparison with CDNN's. For the BCI 2A and Physionet datasets,after graph convolution, we concatenate every channel's full sequence of time points into a single long feature vector which is fed into a simple fully-connected network (MLP), and we apply a Tanh nonlinearity followed by the classification layer.This is done for all graph models. For CDNN in all Datasets we ignore the first un-filtered term (k = 0) to reduce noise.

The financial datasets were z-score normalized and involve the same architecture except we use an ELU activation for all models . The Adam (Kingma & Ba, 2015) optimizer was used in all cases. For the financial data we use a 60/20/20 Train/Val/Test split and for all datasets we evaluate on test data using the model with lowest validation loss during training.

**BCI Competition IV 2a Dataset**

This dataset contains EEG data from nine healthy participants performing a four-class motor imagery task (tongue, feet, right hand, left hand). Signals were recorded from 22 electrodes at 250 Hz. Each participant completed two sessions (training and testing) on separate days, with each session comprising 288 trials of 4 s each. Model evaluation employed leave-one-subject-out cross-validation: for each fold, all trials (both sessions) of one subject were held out as the test set, while the trials of the remaining eight subjects formed the training set.

The raw EEG recordings are first reduced to include only the channels of interest: all non-EEG channels (such as MEG and stimulus lines) are removed, leaving only the EEG signals. Next, the remaining signals are scaled by a factor of $10^6$ to convert from volts to microvolts, ensuring numerical stability and consistency in subsequent processing. A zero-phase Butterworth band-pass filter between 0.01 Hz and 38 Hz is then applied to eliminate slow drifts and high-frequency artifacts. Following filtering, each channel is standardized over time using an exponential moving estimate of its mean and variance, with an adaptation rate $\alpha = 1 \times 10^{-3}$ and an initialization block of 1000 samples; this continuous normalization mitigates changes in signal amplitude and variance across the recording. Finally, fixed-length epochs are extracted around each event marker by applying a trial start offset of $-0.5$ s before cue onset (with no post-cue offset), producing uniform windows ready for feature extraction and classification.

### A.3.1 Learnable $\beta$ Filter Bank

We evaluate a filter bank of three learnable $\beta$ values and observe no degradation in accuracy compared to hand picking $\beta$ values. We note that choosing values beforehand based on domain-knowledge can reduce training cost.

Table 5: Accuracy (%) with learnable $\beta$ filter bank

| Dataset | Accuracy |
|---|---|
| BCI 2A | 49.48 |
| PhysioNet | 80.09 |
| Avg | 64.78 |

### A.4 Ablation Analysis

### A.4.1 EEG

We perform our ablation study on the most challenging dataset, the BCI-2A MI dataset. We repeat the exact analysis over each independent participant using CDNNs with different values of $\beta$ and finally with a filter bank of $\beta$'s. We also compare this to the same model without the covariance-density transform (i.e. VNN). To correct for chance agreement in this multi-class task, we report Cohen's kappa score.

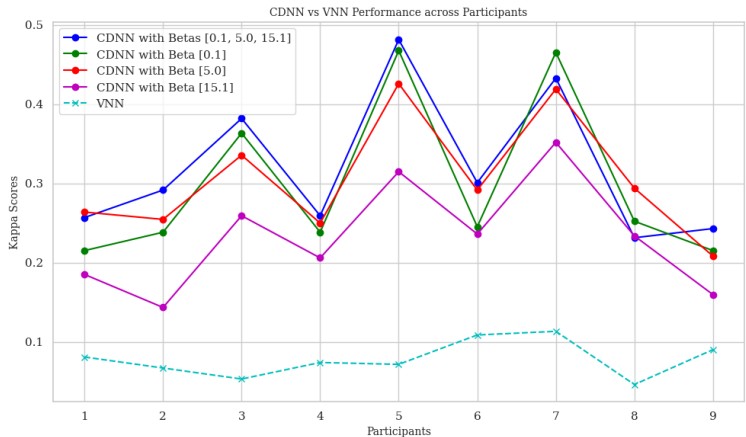

Figure 4: Ablation over different configurations for each subject.

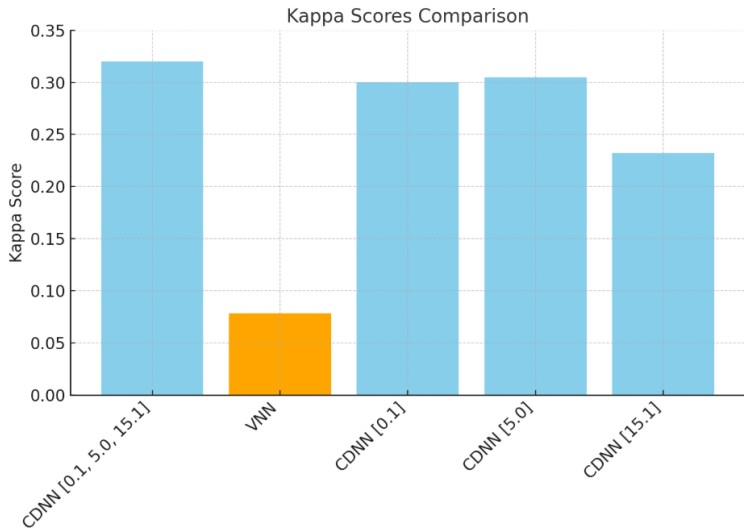

Figure 5: Mean Cohen's kappa for each ablation configuration.

We can clearly see that the covariance-density transform gives us a substantial improvement. Lower $\beta$ values, in general, correspond to higher kappa (noise robustness), yet $\beta = 5$ outperforms $\beta = 0.1$ due to increased discriminability. Very large $\beta$ remains noise-sensitive. Filter banks inherit both benefits, yielding the best performance.

**Financial Forecasting**

We perform an ablation study on the US Stock dataset (horizon $h = 3$) to analyze the role of the diffusion parameter $\beta$.

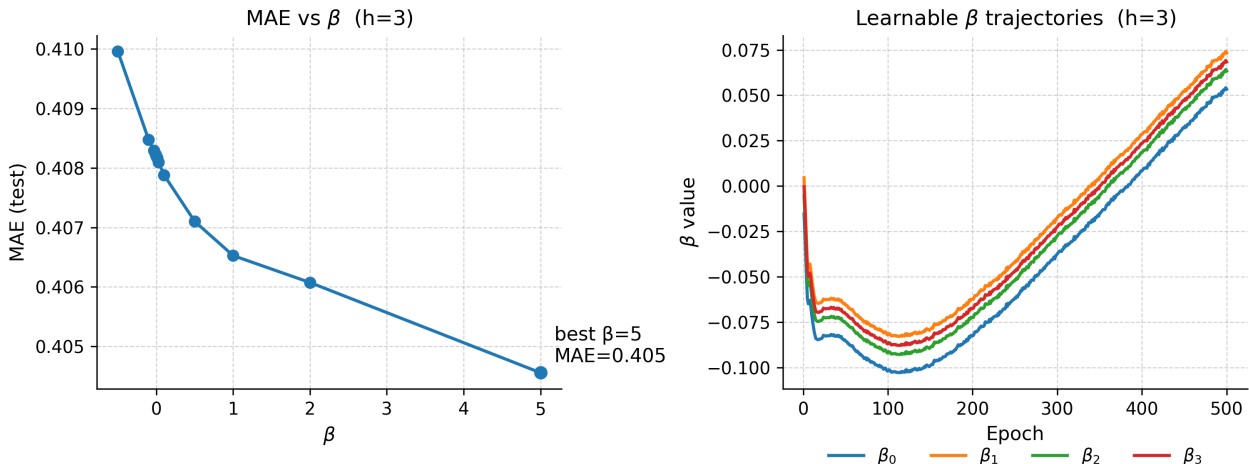

Figure 6: Ablation on the US Stock Exchange Dataset

In the left panel of Fig. 6, we sweep over fixed values of $\beta$ in a single–scale CDNN and report the mean absolute error (MAE) on the US Stock dataset. Performance steadily improves as $\beta$ increases, with the best score obtained at $\beta = 5$. This indicates that stronger diffusion improves performance in this dataset.

In the right panel, we initialize a multi–scale CDNN with four distinct learnable $\beta$ values close to zero and plot their trajectories during training. Interestingly, the $\beta$ values initially drift slightly negative before steadily increasing towards small positive values ($\approx 0.05$–$0.075$ by epoch 500).

## A.5 Scalability Analysis

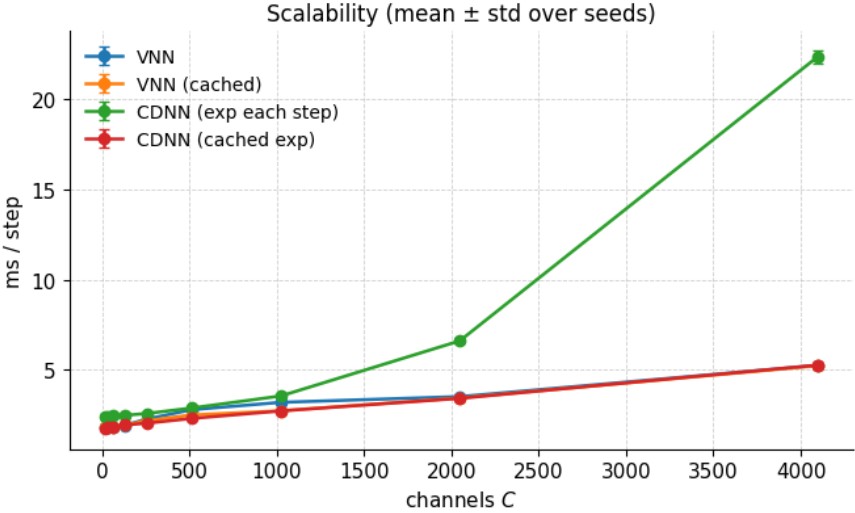

Figure 7: **Runtime scalability of CDNN vs. VNN.** Mean $\pm$ standard deviation of per-step runtime (ms) as a function of the number of channels ($C$). CDNNs scale linearly up to about $C{=}1024$, after which the cost of repeatedly computing the matrix exponential $\exp(-\beta L)$ grows rapidly. When caching the exponential, CDNN performance matches the near-linear scaling of VNNs, confirming that the additional cost can be effectively handled in practice.

We evaluated the computational scalability of CDNNs compared to standard VNNs by measuring the average per-step runtime (forward + backward) across varying numbers of channels ($C$). Four configurations were tested: (1) VNN, (2) VNN with cached Laplacian, (3) CDNN computing $\exp(-\beta L)$ at each step, and

(4) CDNN using a precomputed exponential (*cached exp*). Results are averaged over multiple seeds with standard deviation shown as error bars.

Overall, CDNNs maintain efficient scaling behaviour for moderate graph sizes, and caching $\exp(-\beta L)$ restores runtime similarity with VNNs even for large-scale settings ($C > 1000$). Given that in all our experiments we pre-compute the covariance matrix exponential only once this shows that CDNN's are indeed fairly scalable.

## A.6 Relationship between Covariance and Laplacian Eigenbasis

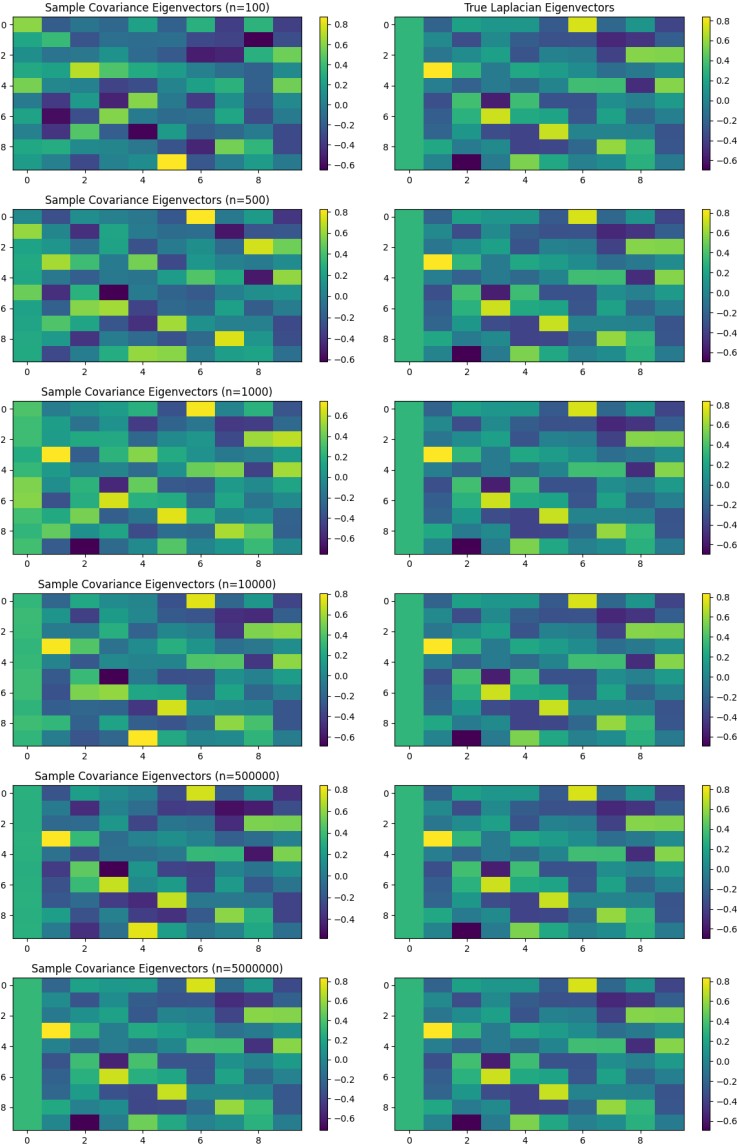

Figure 8: Using signals generated from an Erdős-Rényi Laplacian convolution show convergence of the eigenvectors of the sample covariance matrix to those of the underlying graph Laplacian

Let $\mathbf{L} \in \mathbb{R}^{N \times N}$ be a (combinatorial or normalized) graph Laplacian with eigen-decomposition

$$\mathbf{L} = \mathbf{U}\,\mathbf{\Lambda}\,\mathbf{U}^{\top}, \qquad \mathbf{\Lambda} = \mathrm{diag}(\lambda_1, \ldots, \lambda_N), \tag{5}$$

where the columns of $\mathbf{U}$ are the Laplacian eigenvectors and $0 = \lambda_1 \leq \lambda_2 \leq \cdots \leq \lambda_N$ are the Laplacian eigenvalues. Throughout, $\mathbb{E}[\cdot]$ denotes statistical expectation and $\mathbf{I}$ is the identity matrix.

Following the graph–stationary framework of Dong *et al.* (Dong et al., 2019), assume the graph signal $\mathbf{x}$ is produced by filtering white noise $\mathbf{w} \sim \mathcal{N}(\mathbf{0}, \mathbf{I})$ through an *order-K polynomial graph filter*

$$\mathbf{x} = \sum_{k=0}^{K} a_k \, \mathbf{L}^k \, \mathbf{w} = g(\mathbf{L}) \, \mathbf{w}, \qquad g(z) = \sum_{k=0}^{K} a_k \, z^k. \tag{6}$$

Because $\mathbf{L}$ and $g(\mathbf{L})$ commute, they are simultaneously diagonalizable by $\mathbf{U}$:

$$\mathbf{C} = \mathbb{E}\left[\mathbf{x}\,\mathbf{x}^\top\right] = g(\mathbf{L}) \, \mathbb{E}\left[\mathbf{w}\,\mathbf{w}^\top\right] g(\mathbf{L})^\top = g(\mathbf{L}) \, g(\mathbf{L}) = h(\mathbf{L}), \tag{7}$$

where $h(z) = g(z)^2$. Equation equation 7 implies the covariance matrix $\mathbf{C}$ and Laplacian $\mathbf{L}$ share the same eigenvectors $\mathbf{U}$; their eigenvalues are merely rescaled. (Dong et al., 2019)

If the graph arises as the precision matrix of a GMRF, $\mathbf{x} \sim \mathcal{N}(\mathbf{0}, \mathbf{L}^\dagger)$, then the covariance is the (Moore–Penrose) pseudo-inverse $\mathbf{L}^\dagger$. Therefore

$$\mathbf{L}^\dagger = \mathbf{U} \, \boldsymbol{\Lambda}^\dagger \, \mathbf{U}^\top, \qquad \boldsymbol{\Lambda}^\dagger = \mathrm{diag}\left(0, \lambda_2^{-1}, \dots, \lambda_N^{-1}\right), \tag{8}$$

so the eigenbasis is *exactly* preserved and the eigenvalues are reciprocals (up to the zero mode). Estimating $\mathbf{L}$ under Laplacian constraints is therefore equivalent to learning a sparse inverse-covariance matrix. (Pavez & Ortega, 2016) Conversely, if one has a well-conditioned sample covariance, its inverse supplies a valid Laplacian candidate.

More generally, any graph operator $\mathbf{S}$ that is a *spectral function* of $\mathbf{L}$, i.e. $\mathbf{S} = f(\mathbf{L})$ for some analytic $f$, commutes with $\mathbf{L}$ and shares its eigenvectors:

$$\mathbf{SL} = \mathbf{LS} \implies \mathbf{S} = \mathbf{U} \, f(\boldsymbol{\Lambda}) \, \mathbf{U}^\top. \tag{9}$$

Hence, whenever the true covariance can be expressed (or approximated) as $f(\mathbf{L})$, the sample covariance obtained from sufficiently many realisations *inherits* the Laplacian eigenbasis. Results quantifying the perturbation of eigenvectors under finite-sample noise (e.g. matrix–Bernstein or Davis–Kahan bounds) then show that $\widehat{\mathbf{C}}$ converges to $\mathbf{C}$ not only in Frobenius norm but also in the subspace spanned by $\mathbf{U}$ (Fontan & Altafini, 2021).

**Proposition 4** (Shared eigenbasis of covariance and Laplacian)**.** *Let $\mathbf{L}$ be symmetric and $\mathbf{C} = g(\mathbf{L})$ for some function $g$ with $g(\lambda_i) \geq 0$. Then $\mathbf{C}$ and $\mathbf{L}$ are simultaneously diagonalisable by the same orthonormal matrix $\mathbf{U}$, and $\mathrm{rank}\,\mathbf{C} = |\{\, i \mid g(\lambda_i) > 0 \}|$. Moreover, if $g$ is monotone decreasing, the ordering of variances in $\mathbf{C}$ is the* reverse *of the Laplacian frequency ordering.*

*Proof.* Since $\mathbf{C} = g(\mathbf{L})$ is a polynomial (or analytic) function of $\mathbf{L}$, it commutes with $\mathbf{L}$. Any two real symmetric matrices that commute are simultaneously diagonalisable by an orthogonal matrix; hence they share $\mathbf{U}$. The rank statement follows because $g(\lambda_i) = 0$ if and only if the corresponding eigenvalue of $\mathbf{C}$ vanishes. If $g$ is decreasing, then $\lambda_i < \lambda_j \implies g(\lambda_i) > g(\lambda_j)$, reversing the order. $\square$

When the number of available graph-signal realisations $n$ is much larger than the graph order $N$, the sample covariance $\widehat{\mathbf{C}}_n$ itself becomes a *spectral surrogate* for the Laplacian $\mathbf{L}$. Hence, one can substitute its leading eigenvectors for those of $\mathbf{L}$ and thereby circumvent the combinatorial optimisation required by sparse-Laplacian estimation procedures (Pavez & Ortega, 2016). Moreover, because graph filters implemented in the covariance domain remain diagonal in the shared eigenbasis $\mathbf{U}$, frequency responses may be designed interchangeably on either operator, allowing practitioners to work with whichever spectrum is numerically better conditioned or easier to estimate in a given application. Finally, even when the inverse covariance $\mathbf{C}^{-1}$ fails to satisfy the strict degree-constraints of a Laplacian and is merely positive-semidefinite, its pseudo-inverse $\mathbf{L}^\dagger$ still admits a meaningful *signed-Laplacian* interpretation that preserves the common eigenbasis while relaxing node-degree constraints.

Under broad conditions, stationary graph processes, GMRFs with Laplacian precision, or any model where the covariance is a spectral function of $\mathbf{L}$ the eigenvectors of the sample covariance *converge* to those of the graph Laplacian. Consequently, algorithms that exploit the covariance matrix inherit a graph-spectral interpretation, while graph filters may be implemented directly in the covariance eigenbasis.

**Proof of Theorem 1**

We aim to prove that the density filter frequency response $h(\rho_i)$ satisfies the Lipschitz condition:

$$|h(\rho(\lambda_2) - h(\rho(\lambda_2))| \leq \alpha|\lambda_2 - \lambda_1|$$

for a constant $\alpha$ that depends on the filter parameters. The frequency response is defined as:

$$h(\rho(\lambda_i)) = \sum_{k=0}^{K} \frac{h_k e^{-\beta\lambda k}}{Z^k},$$

where:

$$Z = \sum_{i=1}^{n} e^{-\beta\lambda_i}.$$

Our focus is to bound the difference $|h(\rho(\lambda_2)) - h(\rho(\lambda_1))|$ and derive a meaningful constant $\alpha$.

The difference between $h(\rho(\lambda_2))$ and $h(\rho(\lambda_1))$ is:

$$|h(\rho(\lambda_2)) - h(\rho(\lambda_1))| = \left| \sum_{k=0}^{K} \left( \frac{h_k e^{-\beta\lambda_2 k}}{Z^k} - \frac{h_k e^{-\beta\lambda_1 k}}{Z^k} \right) \right|.$$

Factoring out the common terms, this becomes:

$$|h(\rho(\lambda_2)) - h(\rho(\lambda_1))| = \left| \sum_{k=0}^{K} \frac{h_k}{Z^k} \left( e^{-\beta\lambda_2 k} - e^{-\beta\lambda_1 k} \right) \right|.$$

Using the Mean Value Theorem, we know that for each $k$, there exists a $\xi \in (\lambda_1, \lambda_2)$ such that:

$$e^{-\beta\lambda_2 k} - e^{-\beta\lambda_1 k} = -\beta k(\lambda_2 - \lambda_1)e^{-\beta\xi k}.$$

Substituting this into the expression, we have:

$$|h(\rho(\lambda_2)) - h(\rho(\lambda_1))| = \left| \sum_{k=0}^{K} \frac{h_k}{Z^k} \left( -\beta k(\lambda_2 - \lambda_1)e^{-\beta\xi k} \right) \right|.$$

Using the Triangle inequality:

$$\left| \sum_{k=0}^{K} a_k \right| \leq \sum_{k=0}^{K} |a_k|,$$

we can bound the summation:

$$|h(\rho(\lambda_2)) - h(\rho(\lambda_1))| \leq \sum_{k=0}^{K} \left| \frac{h_k}{Z^k} \left( -\beta k(\lambda_2 - \lambda_1)e^{-\beta\xi k} \right) \right|.$$

Now, factor out $|\lambda_2 - \lambda_1|$:

$$|h(\rho(\lambda_2)) - h(\rho(\lambda_1))| \leq |\lambda_2 - \lambda_1| \sum_{k=0}^{K} \left| \frac{h_k \beta k e^{-\beta\xi k}}{Z^k} \right|.$$

By definition of the partition function, $Z^k$ is at least as large as $e^{-\beta\xi k}$, ensuring that $\frac{e^{-\beta\xi k}}{Z^k} \leq 1$. Using this bound, we simplify:

$$\left| \frac{h_k \beta k e^{-\beta\xi k}}{Z^k} \right| \leq |h_k||\beta k|.$$

Substituting this into the inequality, we get:

$$|h(\rho(\lambda_2)) - h(\rho(\lambda_1))| \leq |\lambda_2 - \lambda_1| \sum_{k=0}^{K} |h_k| |\beta k|.$$

Define the constant $\alpha$ as:

$$\alpha = \sum_{k=0}^{K} |h_k| |\beta k|.$$

Thus, we have:

$$|h(\rho(\lambda_2)) - h(\rho(\lambda_1))| \leq \alpha |\lambda_2 - \lambda_1|.$$

Thus the covariance density filter is Lipchitz Continuous with respect to the denisty transformation, assuming constant filter coefficients.

While stability bounds can be obtained for relative perturbations using Lipschitz filters more illuminating bounds can be obtained using integral Lipschitz filters. It can be shown that under certain conditions the covariance density filter satisfies being integral Lipschitz continuous.

The *Integral Lipschitz condition* imposes the additional requirement:

$$|h(\lambda_2) - h(\lambda_1)| \leq \theta \cdot \frac{|\lambda_2 - \lambda_1|}{\frac{|\lambda_1 + \lambda_2|}{2}}.$$

To satisfy this condition, the constant $\alpha$ must be bounded by:

$$\alpha \leq \frac{1}{\sup\left(\frac{|\lambda_1 + \lambda_2|}{2}\right)},$$

where the supremum is taken over all pairs of eigenvalues $\lambda_1$ and $\lambda_2$.

I.e the following condition must hold:

$$\sum_{k=0}^{K} |h_k| |\beta k| \leq \frac{1}{\sup\left(\frac{|\lambda_1 + \lambda_2|}{2}\right)}.$$

If this condition is satisfied, the filter satisfies the Composite Integral Lipschitz condition with a constant of 1. Further observe The Integral Lipschitz condition can be generalized to allow any real positive constant $\theta > 0$. In this case, the condition becomes:

$$\alpha \leq \frac{\theta}{\sup\left(\frac{|\lambda_1 + \lambda_2|}{2}\right)}.$$

This means that the filter satisfies the Integral Lipschitz condition with a constant $\theta$ as long as $\alpha$ satisfies the above bound.

It follows that as long as $\alpha$ is finite, the filter will satisfy the Integral Lipschitz condition for some choice of *theta*. The constant $\theta$ can be tuned by adjusting the parameters $|h_k|$ (the filter coefficients) and $\beta$.

**Proof of Theorem 2**

We aim to prove that the filter

$$\mathbf{H}(\rho(\mathbf{S})) = \sum_{k=0}^{K} h_k \frac{e^{-\beta \mathbf{S} k}}{(\mathrm{Tr}(e^{-\beta \mathbf{S}}))^k}$$

is permutation equivariant in the vertex domain. Specifically, for a permutation matrix $\mathbf{T}$, we want to show:

$$\mathbf{H}(\rho(\hat{\mathbf{S}}))\hat{\mathbf{x}} = \mathbf{T}^T\mathbf{H}(\rho(\mathbf{S}))\mathbf{x},$$

where $\hat{\mathbf{S}} = \mathbf{T}^T\mathbf{S}\mathbf{T}$ is the permuted graph shift operator and $\hat{\mathbf{x}} = \mathbf{T}^T\mathbf{x}$ is the permuted input signal.

This demonstrates that applying a permutation to the graph and the graph signal results in a permutation of the filter output.

For the permuted graph, the shift operator is $\hat{\mathbf{S}} = \mathbf{T}^T\mathbf{S}\mathbf{T}$. Substituting this into the filter definition, we have:

$$\mathbf{H}(\rho(\hat{\mathbf{S}})) = \sum_{k=0}^{K} h_k \frac{e^{-\beta\hat{\mathbf{S}}k}}{(\mathrm{Tr}(e^{-\beta\hat{\mathbf{S}}}))^k}.$$

Using the Property of matrix exponentials under similarity transformations:

$$e^{-\beta\hat{\mathbf{S}}k} = e^{-\beta(\mathbf{T}^T\mathbf{S}\mathbf{T})k} = \mathbf{T}^T e^{-\beta\mathbf{S}k}\mathbf{T}.$$

The trace of a matrix is invariant under cyclic permutations:

$$(\mathrm{Tr}(e^{-\beta\hat{\mathbf{S}}}))^k = (\mathrm{Tr}(\mathbf{T}^T e^{-\beta\mathbf{S}}\mathbf{T}))^k = (\mathrm{Tr}(e^{-\beta\mathbf{S}}))^k.$$

Substituting these Properties into the filter definition:

$$\mathbf{H}(\rho(\hat{\mathbf{S}})) = \sum_{k=0}^{K} h_k \frac{\mathbf{T}^T e^{-\beta\mathbf{S}k}\mathbf{T}}{(\mathrm{Tr}(e^{-\beta S}))^k}.$$

Factor $\mathbf{T}^T$ and $\mathbf{T}$ outside the summation:

$$H(\hat{\mathbf{S}}) = \mathbf{T}^T \left(\sum_{k=0}^{K} h_k \frac{e^{-\beta\mathbf{S}k}}{(\mathrm{Tr}(e^{-\beta\mathbf{S}}))^k}\right) \mathbf{T} = \mathbf{T}^T\mathbf{H}(\mathbf{S})\mathbf{T}.$$

Let the input signal $\mathbf{x}$ and its permuted version $\hat{\mathbf{x}}$ satisfy $\hat{\mathbf{x}} = \mathbf{T}^T\mathbf{x}$. Applying $\mathbf{H}(\hat{\mathbf{S}})$ to $\hat{\mathbf{x}}$:

$$\hat{\mathbf{z}} = \mathbf{H}(\rho(\hat{\mathbf{S}}))\hat{\mathbf{x}} = \mathbf{T}^T\mathbf{H}(\rho(\mathbf{S}))\mathbf{T}\mathbf{T}^T\mathbf{x}.$$

Using the orthogonality of $\mathbf{P}$, where $\mathbf{T}\mathbf{T}^T = \mathbf{I}$, this simplifies to:

$$\hat{\mathbf{z}} = \mathbf{T}^T H(\mathbf{S})\mathbf{x}.$$

Thus, the filter output for the permuted graph and signal is the permuted version of the original filter output:

$$\hat{\mathbf{z}} = \mathbf{T}^T\mathbf{z},$$

where $\mathbf{z} = \mathbf{H}(\rho(\mathbf{S}))\mathbf{x}$.

**Proof of Lemma 1**

*Proof.* We decompose the error as

$$\mathbf{E} = \frac{e^{-\beta\hat{\mathbf{C}}} - e^{-\beta\mathbf{C}}}{Z'} + e^{-\beta\mathbf{C}}\left(\frac{1}{Z'} - \frac{1}{Z}\right).$$

Taking the operator norm and applying the triangle inequality yields

$$\|\mathbf{E}\| \leq \frac{\|e^{-\beta \hat{\mathbf{C}}} - e^{-\beta \mathbf{C}}\|}{Z'} + \|e^{-\beta \mathbf{C}}\| \cdot \left| \frac{1}{Z'} - \frac{1}{Z} \right|.$$

Next, after an application of Duhamel's formula (Kato, 1966) we have

$$e^{-\beta(\mathbf{C}+\delta\mathbf{C})} - e^{-\beta\mathbf{C}} = -\int_0^\beta e^{-s(\mathbf{C}+\delta\mathbf{C})} \delta\mathbf{C} \, e^{-(\beta-s)\mathbf{C}} \, ds.$$

Changing the variable via $s = \beta t$ (so that $ds = \beta \, dt$, $t \in [0,1]$) yields

$$e^{-\beta(\mathbf{C}+\delta\mathbf{C})} - e^{-\beta\mathbf{C}} = -\beta \int_0^1 e^{-\beta t(\mathbf{C}+\delta\mathbf{C})} \delta\mathbf{C} \, e^{-\beta(1-t)\mathbf{C}} \, dt.$$

Taking the operator norm and using submultiplicativity gives

$$\|e^{-\beta(\mathbf{C}+\delta\mathbf{C})} - e^{-\beta\mathbf{C}}\| \leq |\beta| \, \|\delta\mathbf{C}\| \int_0^1 \|e^{-\beta t(\mathbf{C}+\delta\mathbf{C})}\| \cdot \|e^{-\beta(1-t)\mathbf{C}}\| \, dt.$$

Define

$$I := \int_0^1 \|e^{-\beta t(\mathbf{C}+\delta\mathbf{C})}\| \|e^{-\beta(1-t)\mathbf{C}}\| \, dt.$$

We now consider the two cases:

*(a) If $\beta \geq 0$:* Since $\mathbf{C}$ and $\mathbf{C} + \delta\mathbf{C}$ are positive semidefinite,

$$\|e^{-\beta t(\mathbf{C}+\delta\mathbf{C})}\| \leq 1, \quad \|e^{-\beta(1-t)\mathbf{C}}\| \leq 1, \quad \forall \, t \in [0,1].$$

Hence,

$$I \leq \int_0^1 1 \, dt = 1.$$

*(b) If $\beta < 0$:* Write $\beta = -|\beta|$. Then

$$\|e^{-\beta t(\mathbf{C}+\delta\mathbf{C})}\| = \|e^{|\beta| t(\mathbf{C}+\delta\mathbf{C})}\| \leq e^{|\beta| t \, \|\mathbf{C}+\delta\mathbf{C}\|},$$

and

$$\|e^{-\beta(1-t)\mathbf{C}}\| \leq e^{|\beta|(1-t) \, \|\mathbf{C}\|}.$$

Thus,

$$I \leq \int_0^1 e^{|\beta| t \, \|\mathbf{C}+\delta\mathbf{C}\|} e^{|\beta|(1-t) \, \|\mathbf{C}\|} \, dt.$$

Since

$$|\beta| t \, \|\mathbf{C} + \delta\mathbf{C}\| + |\beta|(1-t) \, \|\mathbf{C}\| = |\beta| \|\mathbf{C}\| + |\beta| t \Big( \|\mathbf{C} + \delta\mathbf{C}\| - \|\mathbf{C}\| \Big),$$

we factor out the constant term:

$$I \leq e^{|\beta| \|\mathbf{C}\|} \int_0^1 \exp\Big\{ |\beta| t \Big( \|\mathbf{C} + \delta\mathbf{C}\| - \|\mathbf{C}\| \Big) \Big\} \, dt.$$

Setting

$$a := |\beta| \Big( \|\mathbf{C} + \delta\mathbf{C}\| - \|\mathbf{C}\| \Big),$$

we have

$$\int_0^1 e^{at} \, dt = \frac{e^a - 1}{a} \, .$$

Thus, for $\beta < 0$

$$I \le e^{|\beta|\|\mathbf{C}\|} \, \frac{e^{|\beta|(\|\mathbf{C}+\delta\mathbf{C}\|-\|\mathbf{C}\|)} - 1}{|\beta|\Big(\|\mathbf{C}+\delta\mathbf{C}\| - \|\mathbf{C}\|\Big)}.$$

We now define the unified factor

$$F(\beta, \mathbf{C}, \delta\mathbf{C}) = \begin{cases} 1, & \beta \ge 0, \\ e^{|\beta|\|\mathbf{C}\|} \, \dfrac{e^{|\beta|(\|\mathbf{C}+\delta\mathbf{C}\|-\|\mathbf{C}\|)} - 1}{|\beta|\Big(\|\mathbf{C}+\delta\mathbf{C}\| - \|\mathbf{C}\|\Big)}, & \beta < 0. \end{cases}$$

To find the limit as $\beta \to 0$:

$$\lim_{\beta \to 0} F(\beta, \mathbf{C}, \delta\mathbf{C})$$

For the Right-Hand Limit$(\beta \to 0^+)$:

$$\lim_{\beta \to 0^+} F(\beta, \mathbf{C}, \delta\mathbf{C}) = 1$$

For the Left-Hand Limit $(\beta \to 0^-)$, Let $\beta = -x$ where $x \to 0^+$:

$$F(-x, \mathbf{C}, \delta\mathbf{C}) = e^{x\|\mathbf{C}\|} \cdot \frac{e^{x(\|\mathbf{C}+\delta\mathbf{C}\|-\|\mathbf{C}\|)} - 1}{x(\|\mathbf{C}+\delta\mathbf{C}\| - \|\mathbf{C}\|)}$$

Let $a = \|\mathbf{C}+\delta\mathbf{C}\| - \|\mathbf{C}\|$:

$$F(-x, \mathbf{C}, \delta\mathbf{C}) = e^{x\|\mathbf{C}\|} \cdot \frac{e^{xa} - 1}{xa}$$

Taking the limit as $x \to 0^+$ a standard result gives us:

$$\lim_{x \to 0^+} e^{x\|\mathbf{C}\|} = 1$$

$$\lim_{x \to 0^+} \frac{e^{xa} - 1}{xa} = 1$$

Therefore:

$$\lim_{\beta \to 0^-} F(\beta, \mathbf{C}, \delta\mathbf{C}) = 1 \times 1 = 1$$

Both the right-hand and left-hand limits as $\beta \to 0$ are equal to 1. Hence,

$$\lim_{\beta \to 0} F(\beta, \mathbf{C}, \delta\mathbf{C}) = 1$$

Continuing,

$$\|e^{-\beta(\mathbf{C}+\delta\mathbf{C})} - e^{-\beta\mathbf{C}}\| \le |\beta| \, \|\delta\mathbf{C}\| \, F(\beta, \mathbf{C}, \delta\mathbf{C}).$$

In particular, setting $\hat{\mathbf{C}} = \mathbf{C} + \delta\mathbf{C}$,

$$\|e^{-\beta\hat{\mathbf{C}}} - e^{-\beta\mathbf{C}}\| \le |\beta| \, \|\delta\mathbf{C}\| \, F(\beta, \mathbf{C}, \delta\mathbf{C}).$$

Hence, the first term in our error decomposition is bounded by

$$\frac{\|e^{-\beta\hat{\mathbf{C}}} - e^{-\beta\mathbf{C}}\|}{Z'} \le \frac{|\beta|\,\|\delta\mathbf{C}\|\,F(\beta, \mathbf{C}, \delta\mathbf{C})}{Z'}.$$

Next, we bound the partition function difference. We have

$$\left|\frac{1}{Z'} - \frac{1}{Z}\right| = \frac{|Z - Z'|}{Z\,Z'}.$$

Since

$$Z' = \mathrm{Tr}\Big(e^{-\beta(\mathbf{C}+\delta\mathbf{C})}\Big) = Z + \mathrm{Tr}\Big(e^{-\beta(\mathbf{C}+\delta\mathbf{C})} - e^{-\beta\mathbf{C}}\Big),$$

and applying $\mathrm{Tr}(A) \le m\,\|A\|$ yields

$$\left|\mathrm{Tr}\Big(e^{-\beta(\mathbf{C}+\delta\mathbf{C})} - e^{-\beta\mathbf{C}}\Big)\right| \le m\,\|e^{-\beta(\mathbf{C}+\delta\mathbf{C})} - e^{-\beta\mathbf{C}}\| \le m\,|\beta|\,\|\delta\mathbf{C}\|\,F(\beta, \mathbf{C}, \delta\mathbf{C}).$$

Thus,

$$|Z - Z'| \le m\,|\beta|\,\|\delta\mathbf{C}\|\,F(\beta, \mathbf{C}, \delta\mathbf{C}).$$

We assume $Z' = R\,Z$ with $R \ge 1$ and $Z \ge 1$ (this holds in high dimensional settings where the covariance matrix is not of full rank and in any case this condition can by regularizing the covariance matrix such that its smallest eigenvalue is 0), i.e the perturbed partition function is some multiple of the true partition function and thus it follows that

$$\left|\frac{1}{Z'} - \frac{1}{Z}\right| \le \frac{m\,|\beta|\,\|\delta\mathbf{C}\|\,F(\beta, \mathbf{C}, \delta\mathbf{C})}{Z\,Z'} \le \frac{m\,|\beta|\,\|\delta\mathbf{C}\|\,F(\beta, \mathbf{C}, \delta\mathbf{C})}{R}.$$

Thus, combining the two parts gives

$$\|\mathbf{E}\| \le \frac{|\beta|\,\|\delta\mathbf{C}\|\,F(\beta, \mathbf{C}, \delta\mathbf{C})}{Z'} + \|e^{-\beta\mathbf{C}}\| \cdot \frac{m\,|\beta|\,\|\delta\mathbf{C}\|\,F(\beta, \mathbf{C}, \delta\mathbf{C})}{R}.$$

Since $Z' = R\,Z$ and $Z \ge 1$ implies $1/Z' \le 1/R$, we obtain

$$\|\mathbf{E}\| \le \frac{|\beta|\,\|\delta\mathbf{C}\|\,F(\beta, \mathbf{C}, \delta\mathbf{C})}{R}\Big(1 + m\,\|e^{-\beta\mathbf{C}}\|\Big).$$

Thus $\forall\beta$

$$m\,\|e^{-\beta\mathbf{C}}\| \le m\,\exp\Big\{\mathbf{1}_{\{\beta<0\}}\,|\beta|\,\|\mathbf{C}\|\Big\}.$$

Finally, the error bound is given as

$$\|\mathbf{E}\| \le \frac{|\beta|\,\|\delta\mathbf{C}\|\,F(\beta, \mathbf{C}, \delta\mathbf{C})}{R}\Big(1 + m\,\exp\Big\{\mathbf{1}_{\{\beta<0\}}\,|\beta|\,\|\mathbf{C}\|\Big\}\Big).$$

$\square$

**Lemma 2** (Perturbation Theory for the Density Matrix). *Consider an ensemble density matrix $\mathbf{P}$ and a sample density matrix $\hat{\mathbf{P}}$. For any eigenvalue $\rho_i > 0$ of $\mathbf{P}$, the perturbation $\mathbf{E}$ satisfies:*

$$\mathbf{E}\,\mathbf{v}_i = \gamma_i\,\delta\mathbf{v}_i + \delta\rho_i\,\mathbf{v}_i + (\delta\rho_i\mathbf{I}_m - \mathbf{E})\,\delta\mathbf{v}_i,$$

*where*

$$\gamma_i = (\rho_i\mathbf{I}_m - \mathbf{P}), \quad \delta\mathbf{v}_i = \mathbf{u}_i - \mathbf{v}_i, \quad \delta\rho_i = w_i - \rho_i.$$

*Proof.* Note this proof follows analogously to the standard result in the perturbations of eigenvalues of the covariance matrix from (Sihag et al., 2022). As the density matrix shares the same eigenbasis as the covariance matrix the proof is straightforward with the only differences being in the transformed eigenvalues.

From the definition of eigenvectors and eigenvalues, we have

$$\hat{\mathbf{P}}\,\mathbf{u}_i = w_i\,\mathbf{u}_i.$$

We rewrite this in terms of perturbations with respect to the ensemble density matrix $\mathbf{P}$ and the outputs of its eigendecomposition as:

$$(\hat{\mathbf{P}} - \mathbf{P})(\mathbf{v}_i + \delta\mathbf{v}_i) + \mathbf{P}(\mathbf{v}_i + \delta\mathbf{v}_i) = (\rho_i + \delta\rho_i)(\mathbf{v}_i + \delta\mathbf{v}_i),$$

where $w_i = \rho_i + \delta\rho_i$ and $\mathbf{u}_i = \mathbf{v}_i + \delta\mathbf{v}_i$.

Using the fact that $\mathbf{P}\,\mathbf{v}_i = \rho_i\,\mathbf{v}_i$ and rearranging terms, we have:

$$(\hat{\mathbf{P}} - \mathbf{P})\mathbf{v}_i = (\rho_i\mathbf{I}_m - \mathbf{P})\delta\mathbf{v}_i + \delta\rho_i(\mathbf{v}_i + \delta\mathbf{v}_i) - (\hat{\mathbf{P}} - \mathbf{P})\delta\mathbf{v}_i.$$

By setting $\mathbf{E} = \hat{\mathbf{P}} - \mathbf{P}$ and $\gamma_i = \rho_i\mathbf{I}_m - \mathbf{P}$, this simplifies to:

$$\mathbf{E}\,\mathbf{v}_i = \gamma_i\,\delta\mathbf{v}_i + \delta\rho_i\,\mathbf{v}_i + (\delta\rho_i\mathbf{I}_m - \mathbf{E})\,\delta\mathbf{v}_i.$$

$\square$

**Proof of Theorem 3**

We note that the density filters with respect to $\hat{\mathbf{P}}$ and $\mathbf{P}$ are given by

$$\mathbf{H}(\hat{\mathbf{P}}) = \sum_{k=0}^{m} h_k\hat{\mathbf{P}}^k \quad \text{and} \quad \mathbf{H}(\mathbf{P}) = \sum_{k=0}^{m} h_k\mathbf{P}^k. \tag{1}$$

We aim to study the stability of the density filters by analyzing the difference between $\mathbf{H}(\hat{\mathbf{P}})$ and $\mathbf{H}(\mathbf{P})$. For this purpose, we next establish the first-order approximation for $\hat{\mathbf{P}}^k$ in terms of $\mathbf{P}$ and $\mathbf{E}$. Using $\hat{\mathbf{P}} = \mathbf{P} + \mathbf{E}$, the first-order approximation of $\hat{\mathbf{P}}^k$ is given by

$$(\mathbf{P} + \mathbf{E})^k = \mathbf{P}^k + \sum_{r=0}^{k-1} \mathbf{P}^r\mathbf{E}\,\mathbf{P}^{k-r-1} + \tilde{\mathbf{E}}, \tag{2}$$

where $\tilde{\mathbf{E}}$ satisfies $\|\tilde{\mathbf{E}}\| \leq \sum_{r=2}^{k} \binom{k}{r}\|\mathbf{E}\|^r\|\mathbf{P}\|^{k-r}$. Using (2), we have

$$\mathbf{H}(\hat{\mathbf{P}}) - \mathbf{H}(\mathbf{P}) = \sum_{k=0}^{m} h_k\left[(\mathbf{P} + \mathbf{E})^k - \mathbf{P}^k\right], \tag{3}$$

$$= \sum_{k=0}^{m} h_k \sum_{r=0}^{k-1} \mathbf{P}^r\mathbf{E}\,\mathbf{P}^{k-r-1} + \tilde{\mathbf{D}}, \tag{4}$$

where $\tilde{\mathbf{D}}$ satisfies $\|\tilde{\mathbf{D}}\|_2 = O(\|\mathbf{E}\|_2)$. The focus of our subsequent analysis will be the first term in (4). For a random data sample $\mathbf{x} = [x_1, \ldots, x_m]^T$, such that $\|\mathbf{x}\| < R$, for some $R > 0$ and $\mathbf{x} \in \mathbb{R}^{m \times 1}$, its VFT with respect to $\mathbf{P}$ is given by $\tilde{\mathbf{x}} = \mathbf{V}^T\mathbf{x}$, where $\tilde{\mathbf{x}} = [\tilde{x}_1, \ldots, \tilde{x}_m]^T$. The relationship between $\tilde{\mathbf{x}}$ and $\mathbf{x}$ can be expressed as

$$\mathbf{x} = \sum_{i=1}^{m} \tilde{x}_i\mathbf{v}_i. \tag{5}$$

Multiplying both sides of (4) by $\mathbf{x}$ and leveraging (5), we get

$$[\mathbf{H}(\hat{\mathbf{P}}) - \mathbf{H}(\mathbf{P})]\mathbf{x} = \sum_{k=0}^{m} h_k \sum_{r=0}^{k-1} \mathbf{P}^r \mathbf{E} \mathbf{P}^{k-r-1} \mathbf{x} + \mathbf{D}\,\tilde{\mathbf{x}}, \tag{6}$$

$$= \sum_{i=1}^{m} \tilde{x}_i \sum_{k=0}^{m} h_k \sum_{r=0}^{k-1} \mathbf{P}^r \rho_i^{k-r-1} \mathbf{E}\,\mathbf{v}_i + \mathbf{D}\,\tilde{\mathbf{x}}, \tag{7}$$

$$= \sum_{i=1}^{m} \tilde{x}_i \sum_{k=0}^{m} h_k \sum_{r=0}^{k-1} \mathbf{P}^r \rho_i^{k-r-1} \gamma_i\,\delta\mathbf{v}_i + \sum_{i=1}^{m} \tilde{x}_i \sum_{k=0}^{m} h_k \sum_{r=0}^{k-1} \mathbf{P}^r \rho_i^{k-r-1} \delta\rho_i\,\mathbf{v}_i + \sum_{i=1}^{m} \tilde{x}_i \sum_{k=0}^{m} h_k \sum_{r=0}^{k-1} \mathbf{P}^r \rho_i^{k-r-1} (\delta\rho_i \mathbf{I}_m - \mathbf{E})\,\delta\mathbf{v}_i. \tag{8}$$

Where we leveraged the result from Lemma 1 that expands $\mathbf{E}\,\mathbf{v}_i$.

**Term 1:**

This follows very similarly from (Sihag et al., 2022). Using $\gamma_i = \rho_i \mathbf{I}_m - \mathbf{P}$ and $\delta\mathbf{v}_i = \mathbf{u}_i - \mathbf{v}_i$ in the first term in (39), we have

$$\sum_{i=1}^{m} \tilde{x}_i \sum_{k=0}^{m} h_k \sum_{r=0}^{k-1} \mathbf{P}^r \rho_i^{k-r-1} (\rho_i \mathbf{I}_m - \mathbf{P})(\mathbf{u}_i - \mathbf{v}_i). \tag{9}$$

Finally, using $\mathbf{P}^r = \mathbf{V}\mathbf{\Lambda}^r\mathbf{V}^T$ in (9), the first term in (8) is equivalent to:

$$\sum_{i=1}^{m} \tilde{x}_i \sum_{k=0}^{m} h_k \sum_{r=0}^{k-1} \rho_i^{k-r-1} \mathbf{V}\mathbf{\Lambda}^r (\rho_i \mathbf{I}_m - \mathbf{\Lambda})\mathbf{V}^T (\mathbf{u}_i - \mathbf{v}_i), \tag{10}$$

$$= \sum_{i=1}^{m} \tilde{x}_i\,\mathbf{V}\mathbf{L}_i\mathbf{V}^T (\mathbf{u}_i - \mathbf{v}_i), \tag{11}$$

where $\mathbf{L}_i$ is a diagonal matrix whose $j$-th element is given by:

$$[\mathbf{L}_i]_j = \sum_{k=0}^{m} h_k \sum_{r=0}^{k-1} (\rho_i - \rho_j)\rho_i^{k-r-1}\rho_j^r, \tag{12}$$

$$= \sum_{k=0}^{m} h_k(\rho_i - \rho_j)\frac{\rho_i^k - \rho_j^k}{\rho_i - \rho_j}, \tag{13}$$

$$= \sum_{k=0}^{m} h_k\rho_i^k - \sum_{k=0}^{m} h_k\rho_j^k, \tag{14}$$

$$= h(\rho_i) - h(\rho_j). \tag{15}$$

After applying the operator norm:

$$\left\| \sum_{i=1}^{m} \tilde{x}_i \sum_{k=0}^{m} h_k \sum_{r=0}^{k-1} \mathbf{P}^r \rho_i^{k-r-1} \gamma_i\,\delta\mathbf{v}_i \right\| \leq \sqrt{m} \sum_{i=1}^{m} |\tilde{x}_i| \max_{j, i \neq j} |h(\rho_i) - h(\rho_j)| \|\mathbf{v}_j^T \mathbf{u}_i\|.$$

Here, $\mathbf{v}_j^T \mathbf{u}_i$ is the inner product between the eigenvector $\mathbf{v}_j$ of the ensemble density matrix $\mathbf{P}$ and the eigenvector $\mathbf{u}_i$ of the sample density matrix $\hat{\mathbf{P}}$.

Using the result from [(Vershynin, 2018), Theorem 4.1 and (lou, 2017)], we conclude that if

$$\text{sgn}(\lambda_j - \lambda_i)^2 w_j > \text{sgn}(\lambda_j - \lambda_i)(\lambda_j - \lambda_i),$$

for $\lambda_i \neq \lambda_j$, the following condition holds:

$$\left\| \sum_{i=1}^{m} \tilde{x}_i \sum_{k=0}^{m} h_k \sum_{r=0}^{k-1} \mathbf{P}^r \rho_i^{k-r-1} \gamma_i \, \delta \mathbf{v}_i \right\| \leq \sqrt{m} \sum_{i=1}^{m} |\tilde{x}_i| \max_{j,i \neq j} \frac{|h(\rho_i) - h(\rho_j)|}{|\lambda_i - \lambda_j|} \frac{2k_i}{N^{1/2-\varepsilon}},$$

with probability at least

$$1 - \frac{1}{N^{2\varepsilon}},$$

for some $\varepsilon \in (0, 1/2]$, where

$$k_i = \sqrt{\mathbb{E}\left[\|\mathbf{X}\mathbf{X}^T \mathbf{v}_i\|_2^2\right] - \rho_i^2}.$$

From Theorem 1 we know that

$$|h(\rho_2) - h(\rho_1)| \leq \alpha |\lambda_2 - \lambda_1|, \quad \alpha = \sum_{k=0}^{K} |h_k||\beta k|,$$

Note now that we can always choose $\alpha$ (i.e by adjusting $\beta$) such that:

$$|h(\rho_i) - h(\rho_j)| \leq \frac{\alpha}{k_i} |\lambda_i - \lambda_j|.$$

Substituting this bound into the above condition and applying a union bound, we have:

$$\left\| \sum_{i=1}^{m} \tilde{x}_i \sum_{k=0}^{m} h_k \sum_{r=0}^{k-1} \mathbf{P}^r \rho_i^{k-r-1} \gamma_i \, \delta \mathbf{v}_i \right\| \leq \frac{2\sum_{k=0}^{K} |h_k||\beta k|}{N^{1/2-\varepsilon}} \sum_{i=1}^{m} |\tilde{x}_i|,$$

with probability at least

$$1 - \frac{1}{N^{2\varepsilon}} - \frac{2\kappa m}{N},$$

where $\kappa$ is as defined in [(Vershynin, 2018), Corollary 4.2 and (Sihag et al., 2022)].

Furthermore, note that $\sqrt{m} \sum_{i=1}^{m} |\tilde{x}_i| \leq m\|\mathbf{x}\|_2$. If the random sample $\mathbf{x}$ satisfies $\|\mathbf{x}\|_2 \leq Q$, then

$$\mathbb{P}\left( \left\| \sum_{i=1}^{m} \tilde{x}_i \sum_{k=0}^{m} h_k \sum_{r=0}^{k-1} \mathbf{P}^r \rho_i^{k-r-1} \gamma_i \, \delta \mathbf{v}_i \right\| \leq \frac{2\sum_{k=0}^{K} |h_k||\beta k|mQ}{N^{1/2-\varepsilon}} \right) \geq 1 - \frac{1}{N^{2\varepsilon}} - \frac{2\kappa m}{N}.$$

**Term 2:**

We start with:

$$\sum_{i=1}^{m} \tilde{x}_i \sum_{k=0}^{m} h_k \sum_{r=0}^{k-1} \mathbf{P}^r \rho_i^{k-r-1} \delta \rho_i \, \mathbf{v}_i = \sum_{i=1}^{m} \tilde{x}_i \sum_{k=0}^{m} h_k k \rho_i^{k-1} \delta \rho_i \, \mathbf{v}_i.$$

Since $\rho_i^{k-1} \leq 1$ for all $\rho_i$ (as eigenvalues of $\mathbf{P}$ lie within the unit range of the covariance density matrix), we simplify:

$$\sum_{i=1}^{m} \tilde{x}_i \sum_{k=0}^{m} h_k k \rho_i^{k-1} \delta \rho_i \, \mathbf{v}_i \leq \sum_{i=1}^{m} \tilde{x}_i \sum_{k=0}^{m} |h_k| k \delta \rho_i \, \mathbf{v}_i.$$

Using the triangle inequality, we relate $\sum_{k=0}^{m} |h_k||k|$ to the constant $\alpha$ from Theorem 1, where:

$$\sum_{k=0}^{m} |h_k||k| \leq \frac{\alpha}{|\beta|}.$$

Substituting this bound, we get:

$$\sum_{i=1}^{m} \tilde{x}_i \sum_{k=0}^{m} h_k k \delta\rho_i \, \mathbf{v}_i \leq \frac{\alpha}{|\beta|} \sum_{i=1}^{m} |\tilde{x}_i| |\delta\rho_i| \|\mathbf{v}_i\|.$$

Since $\|\mathbf{v}_i\| = 1$, this simplifies further to:

$$\sum_{i=1}^{m} \tilde{x}_i \sum_{k=0}^{m} h_k k \delta\rho_i \, \mathbf{v}_i \leq \frac{\alpha}{|\beta|} \sum_{i=1}^{m} |\tilde{x}_i| |\delta\rho_i|.$$

From Lemma 2, we have the bound

$$\|\mathbf{E}\| \leq \frac{|\beta| \, \|\delta\mathbf{C}\| \, F(\beta, \mathbf{C}, \delta\mathbf{C})}{R} \Big( 1 + m \, \exp\Big\{ \mathbf{1}_{\{\beta<0\}} |\beta| \, \|\mathbf{C}\| \Big\} \Big).$$

By Weyl's theorem, $|\delta\rho_i| \leq \|\mathbf{E}\|$ (Golub & Loan, 2013), so:

$$|\delta\rho_i| \leq \frac{|\beta| \, \|\delta\mathbf{C}\| \, F(\beta, \mathbf{C}, \delta\mathbf{C})}{R} \Big( 1 + m \, \exp\Big\{ \mathbf{1}_{\{\beta<0\}} |\beta| \, \|\mathbf{C}\| \Big\} \Big).$$

Substituting this bound into our inequality:

$$\sum_{i=1}^{m} \tilde{x}_i \sum_{k=0}^{m} h_k k \delta\rho_i \, \mathbf{v}_i \leq \frac{\alpha}{|\beta|} \sum_{i=1}^{m} |\tilde{x}_i| \frac{|\beta| \, \|\delta\mathbf{C}\| \, F(\beta, \mathbf{C}, \delta\mathbf{C})}{R} \Big( 1 + m \, \exp\Big\{ \mathbf{1}_{\{\beta<0\}} |\beta| \, \|\mathbf{C}\| \Big\} \Big).$$

Canceling the $|\beta|$ terms, we obtain

$$\sum_{i=1}^{m} \tilde{x}_i \sum_{k=0}^{m} h_k \, k \, \delta\rho_i \, \mathbf{v}_i \leq \frac{\alpha \|\delta\mathbf{C}\| \, F(\beta, \mathbf{C}, \delta\mathbf{C})}{R} \Big( 1 + m \, \exp\Big\{ \mathbf{1}_{\{\beta<0\}} |\beta| \, \|\mathbf{C}\| \Big\} \Big) \sum_{i=1}^{m} |\tilde{x}_i|.$$

Finally, noting that $\sum_{i=1}^{m} |\tilde{x}_i| \leq \sqrt{m} \|\mathbf{x}\|_2$ and if $\|\mathbf{x}\|_2 \leq Q$, we have:

$$\sum_{i=1}^{m} \tilde{x}_i \sum_{k=0}^{m} h_k \, k \, \delta\rho_i \, \mathbf{v}_i \leq \frac{\alpha \|\delta\mathbf{C}\| \, F(\beta, \mathbf{C}, \delta\mathbf{C})}{R} \Big( 1 + m \, \exp\Big\{ \mathbf{1}_{\{\beta<0\}} |\beta| \, \|\mathbf{C}\| \Big\} \Big) \sqrt{m} Q.$$

**Term 3:** With the same argument regarding the invariance to shifts in eigenbasis it follows from [3] that:

$$\|\delta\rho_i \mathbf{I}_m - \mathbf{E}\| \leq 2 \, \|\mathbf{E}\|,$$

$$\|\delta\mathbf{v}_i\| = \mathcal{O}\left( \frac{1}{\sqrt{N}} \right) \quad \text{with high probability.}$$

Furthermore using the fact that for a random instance $\mathbf{x}$ of random vector $\mathbf{X}$ whose probability distribution is supported within a bounded region w.l.o.g, such that $\|\mathbf{x}\| \leq 1$, for some constant $B > 0$ and $u > 0$, we have

$$\mathbb{P}\left( \|\delta\mathbf{C}\| \leq B \|\mathbf{C}\| \sqrt{\frac{\log m + u}{N}} + \big(1 + \|\mathbf{C}\|\big) \frac{\log m + u}{N} \right) \geq 1 - 2^{-u},$$

We can expand out Lemma 1 as:

$$\|\mathbf{E}\| \le \frac{|\beta|}{R} \, \|\delta\mathbf{C}\| \, F(\beta, \mathbf{C}, \delta\mathbf{C}) \left( 1 + m \exp\left[ 1_{\{\beta<0\}} \, |\beta| \, \|\mathbf{C}\| \right] \right)$$

$$\le \frac{|\beta|}{R} \left[ B \, \|\mathbf{C}\| \sqrt{\frac{\log m + u}{N}} + (1 + \|\mathbf{C}\|) \frac{\log m + u}{N} \right] F(\beta, \mathbf{C}, \delta\mathbf{C}) \, R \left( 1 + m \exp\left[ n \, 1_{\{\beta<0\}} \, |\beta| \, \|\mathbf{C}\| \right] \right)$$

$$= \frac{|\beta|}{R} \, B \, \|\mathbf{C}\| \sqrt{\frac{\log m + u}{N}} F(\beta, \mathbf{C}, \delta\mathbf{C}) \left( 1 + m \exp\left[ 1_{\{\beta<0\}} \, |\beta| \, \|\mathbf{C}\| \right] \right)$$

$$+ \frac{|\beta|}{R} \left( 1 + \|\mathbf{C}\| \right) \frac{\log m + u}{N} F(\beta, \mathbf{C}, \delta\mathbf{C}) \left( 1 + m \exp\left[ 1_{\{\beta<0\}} \, |\beta| \, \|\mathbf{C}\| \right] \right).$$

Thus:

$$\|\mathbf{E}\| = \mathcal{O}\left( \frac{1}{\sqrt{N}} \right) \text{ with high probability.}$$

Which in turns implies:

$$(\delta\rho_i \mathbf{I}_m - \mathbf{E}) \, \delta\mathbf{v}_i = \mathcal{O}\left( \frac{1}{N} \right),$$

Note that for positive $\beta$ this always holds and for negative $\beta$ since from Lemma 1 $F(\beta, \mathbf{C}, \delta\mathbf{C})$ tends to 1 as $\beta$ tends to 0 we can always pick small $\beta$ to ignore the $F$ term. Thus term 3 diminishes faster with $N$ as compared to terms 1 and terms 2 and thus terms 1 and 2 dominate the scaling behaviour of the overall upper bound.

The overall proof is completed by noting that the condition on $\|[H(\hat{\mathbf{P}}) - H(\mathbf{P})]\mathbf{x}\|$ simplifies to the condition on the operator norm $\|[H(\hat{\mathbf{P}}) - H(\mathbf{P})]\|$ for any $\|\mathbf{x}\| \le 1$.

By unrolling this bound through $L$ layers, and noting that each layer can at most amplify the perturbation by a factor of $F$, we obtain the overall network stability bound. The factor $LF^{L-1}$ appears naturally from composing $L$ layers each with at most $F$-fold channel combination where for each $i \in F$ there are potentially different $\alpha_i$ and $\beta_i$. The filter-level bound then carries through all layers.

Since the probability bounds and constants are inherited from the filter-level analysis, the final network-level bound follows directly, concluding the proof.

## A.7 Sub-Additivity for Multi-Scale Von Neumann Entropy for Covariance Matrices

For this definition of entropy for covariance matrices to hold we would like to achieve the desirable sub-additivity Property of a valid entropy measure. The following theorem shows that this Property does indeed hold.

**Theorem 5.** *Let* $\mathbf{C}^1, \dots, \mathbf{C}^n \in \mathbb{R}^{N \times N}$ *be Individual Covariance Matrices. Define*

$$\mathbf{\Sigma}_n \; = \; \sum_{j=1}^{n} \mathbf{C}^j.$$

*For each matrix* $\mathbf{X}$*, let*

$$\rho_\mathbf{X} \; = \; \frac{e^{-\beta\mathbf{X}}}{\mathrm{Tr}[e^{-\beta\mathbf{X}}]}, \quad Z_\mathbf{X} \; = \; \mathrm{Tr}[e^{-\beta\mathbf{X}}], \quad S(\mathbf{X}) \; = \; \beta \, \mathrm{Tr}[\mathbf{X} \, \rho_\mathbf{X}] \; + \; \ln(Z_\mathbf{X}).$$

*Then*

$$S\left(\boldsymbol{\Sigma}_n = \sum_{j=1}^{n} \mathbf{C}^j\right) \;\leq\; \sum_{j=1}^{n} S(\mathbf{C}^j).$$

*I.e. the Von-Neumann Entropy for Covariance Matrices satisfies the sub-additivity Property.*

*Proof.* The main chunk of this proof is the same as the one provided by Domenico et al. (Domenico & Biamonte, 2016) for the Laplacian-based density matrix, however we use a regularization trick to ensure that the partition function of the covariance density matrix is always $\geq 1$, ensuring non-negativity.

We will prove this by induction:

**Base case** ($n = 2$)**.** We first prove the result for two covariance matrices $\mathbf{C}^1$ and $\mathbf{C}^2$. Let us set $\boldsymbol{\Sigma}_2 := \mathbf{C}^1 + \mathbf{C}^2$. We want to show

$$S\left(\mathbf{C}^1 + \mathbf{C}^2\right) \;\leq\; S(\mathbf{C}^1) + S(\mathbf{C}^2).$$

We first regularize each matrix so that $\min \lambda = 0$. For $j = 1, 2$, define

$$m^j \;=\; \min\{\lambda \mid \lambda \text{ is an eigenvalue of } \mathbf{C}^j\} \quad (\geq 0 \text{ since } \mathbf{C}^j \text{ is p.s.d.}).$$

Let

$$\widetilde{\mathbf{C}}^j \;=\; \mathbf{C}^j - m^j \mathbf{I}.$$

Then $\min \lambda(\widetilde{\mathbf{C}}^j) = 0$.

Observe:

$$S(\widetilde{\mathbf{C}}^j) \;=\; S(\mathbf{C}^j),$$

because shifting by $m^j \mathbf{I}$ only multiplies $e^{-\beta \mathbf{C}^j}$ by a factor $e^{+\beta m^j}$ which cancels in the normalized density matrix. Concretely,

$$e^{-\beta(\widetilde{\mathbf{C}}^j)} = e^{+\beta m^j} e^{-\beta \mathbf{C}^j} \implies \rho_{\widetilde{\mathbf{C}}^j} = \frac{e^{-\beta(\widetilde{\mathbf{C}}^j)}}{\mathrm{Tr}\left[e^{-\beta(\widetilde{\mathbf{C}}^j)}\right]} = \frac{e^{+\beta m^j} e^{-\beta \mathbf{C}^j}}{e^{+\beta m^j} \mathrm{Tr}\left[e^{-\beta \mathbf{C}^j}\right]} = \rho_{\mathbf{C}^j}, \;\; S(\widetilde{\mathbf{C}}^j) = S(\mathbf{C}^j).$$

We define

$$\widetilde{\boldsymbol{\Sigma}}_2 \;=\; \widetilde{\mathbf{C}}^1 + \widetilde{\mathbf{C}}^2.$$

We rename:

$$\mathbf{A} := \widetilde{\mathbf{C}}^1, \quad \mathbf{B} := \widetilde{\mathbf{C}}^2, \quad \mathbf{C} := \mathbf{A} + \mathbf{B}.$$

All three $\mathbf{A}, \mathbf{B}, \mathbf{C}$ now have $\min(\lambda) = 0$. Also

$$S(\mathbf{A}) = S(\mathbf{C}^1), \quad S(\mathbf{B}) = S(\mathbf{C}^2), \quad S(\mathbf{A} + \mathbf{B}) = S(\mathbf{C}^1 + \mathbf{C}^2).$$

Define

$$\rho_{\mathbf{A}} = \frac{e^{-\beta \mathbf{A}}}{Z_{\mathbf{A}}}, \quad \rho_{\mathbf{B}} = \frac{e^{-\beta \mathbf{B}}}{Z_{\mathbf{B}}}, \quad \rho_{\mathbf{A}+\mathbf{B}} = \frac{e^{-\beta(\mathbf{A}+\mathbf{B})}}{Z_{\mathbf{A}+\mathbf{B}}},$$

where $Z_{\mathbf{X}} = \mathrm{Tr}\left[e^{-\beta \mathbf{X}}\right]$ and $S(\mathbf{X}) = \beta \mathrm{Tr}[\mathbf{X} \rho_{\mathbf{X}}] + \ln(Z_{\mathbf{X}})$. Because $\mathbf{A}, \mathbf{B}$ each has a zero eigenvalue (or rank deficiency), $Z_{\mathbf{A}}, Z_{\mathbf{B}}, Z_{\mathbf{A}+\mathbf{B}} \geq 1$.

We now consider two KL divergences:

$$D\left(\rho_{\mathbf{A}+\mathbf{B}} \,\|\, \rho_{\mathbf{A}}\right) = \mathrm{Tr}\left[\rho_{\mathbf{A}+\mathbf{B}} \left(\ln \rho_{\mathbf{A}+\mathbf{B}} - \ln \rho_{\mathbf{A}}\right)\right] \;\geq 0,$$

$$D\left(\rho_{\mathbf{A}+\mathbf{B}} \,\|\, \rho_{\mathbf{B}}\right) \;\geq 0.$$

Expanding each yields:

Hence
$$\ln \rho_{\mathbf{A+B}} = -\beta(\mathbf{A+B}) - \ln Z_{\mathbf{A+B}}, \quad \ln \rho_{\mathbf{A}} = -\beta\mathbf{A} - \ln Z_{\mathbf{A}}.$$

So
$$\ln \rho_{\mathbf{A+B}} - \ln \rho_{\mathbf{A}} = -\beta\big[(\mathbf{A+B}) - \mathbf{A}\big] - \ln Z_{\mathbf{A+B}} + \ln Z_{\mathbf{A}} = -\beta\mathbf{B} - \ln Z_{\mathbf{A+B}} + \ln Z_{\mathbf{A}}.$$

Thus
$$D\big(\rho_{\mathbf{A+B}}\|\rho_{\mathbf{A}}\big) = \mathrm{Tr}\Big[\rho_{\mathbf{A+B}}\big(\ln \rho_{\mathbf{A+B}} - \ln \rho_{\mathbf{A}}\big)\Big] = \mathrm{Tr}\Big[\rho_{\mathbf{A+B}}\big(-\beta\mathbf{B} - \ln Z_{\mathbf{A+B}} + \ln Z_{\mathbf{A}}\big)\Big].$$

Because $\mathrm{Tr}[\rho_{\mathbf{A+B}}] = 1$,
$$D\big(\rho_{\mathbf{A+B}}\|\rho_{\mathbf{A}}\big) = -\beta\,\mathrm{Tr}[\mathbf{B}\,\rho_{\mathbf{A+B}}] - \ln Z_{\mathbf{A+B}} + \ln Z_{\mathbf{A}}.$$

Meanwhile $S(\mathbf{A+B}) = \beta\,\mathrm{Tr}[(\mathbf{A+B})\,\rho_{\mathbf{A+B}}] + \ln Z_{\mathbf{A+B}}$. Observe
$$-\beta\,\mathrm{Tr}[\mathbf{B}\,\rho_{\mathbf{A+B}}] - \ln Z_{\mathbf{A+B}} = -\beta\,\mathrm{Tr}[(\mathbf{A+B})\,\rho_{\mathbf{A+B}}] + \beta\,\mathrm{Tr}[\mathbf{A}\,\rho_{\mathbf{A+B}}] - \ln Z_{\mathbf{A+B}} = -S(\mathbf{A+B}) + \beta\,\mathrm{Tr}[\mathbf{A}\,\rho_{\mathbf{A+B}}].$$

Hence
$$D\big(\rho_{\mathbf{A+B}}\|\rho_{\mathbf{A}}\big) = -S(\mathbf{A+B}) + \beta\,\mathrm{Tr}[\mathbf{A}\,\rho_{\mathbf{A+B}}] + \ln Z_{\mathbf{A}}.$$

Since $D(\rho_{\mathbf{A+B}}\|\rho_{\mathbf{A}}) \geq 0$ we get
$$-S(\mathbf{A+B}) + \beta\,\mathrm{Tr}[\mathbf{A}\,\rho_{\mathbf{A+B}}] + \ln Z_{\mathbf{A}} \;\geq\; 0. \tag{1}$$

Likewise,
$$\rho_{\mathbf{B}} = e^{-\beta\mathbf{B}}/Z_{\mathbf{B}}, \quad D\big(\rho_{\mathbf{A+B}}\|\rho_{\mathbf{B}}\big) = \mathrm{Tr}\big[\rho_{\mathbf{A+B}}\,(\ln \rho_{\mathbf{A+B}} - \ln \rho_{\mathbf{B}})\big] \geq 0.$$

One finds (by the same step):
$$D\big(\rho_{\mathbf{A+B}}\|\rho_{\mathbf{B}}\big) = -S(\mathbf{A+B}) + \beta\,\mathrm{Tr}[\mathbf{B}\,\rho_{\mathbf{A+B}}] + \ln Z_{\mathbf{B}} \;\geq 0. \tag{2}$$

We now have:
$$(1)\colon \quad -S(\mathbf{A+B}) + \beta\,\mathrm{Tr}[\mathbf{A}\,\rho_{\mathbf{A+B}}] + \ln Z_{\mathbf{A}} \;\geq 0,$$
$$(2)\colon \quad -S(\mathbf{A+B}) + \beta\,\mathrm{Tr}[\mathbf{B}\,\rho_{\mathbf{A+B}}] + \ln Z_{\mathbf{B}} \;\geq 0.$$

Further consider the fact that covariance matrices and their covariance density counterparts are always positive semi-definite. Therefore, the terms
$$(3)\colon \quad \mathrm{Tr}[\mathbf{A}\,\rho_{\mathbf{A}}] \geq 0,$$
$$(4)\colon \quad \mathrm{Tr}[\mathbf{B}\,\rho_{\mathbf{B}}] \geq 0$$

are always non-negative.

Since at least one eigenvalue of the covariance matrix is 0 due to the regularization trick (and thus their sum), the term
$$(5)\colon \quad \ln Z_{\mathbf{C}}$$
is also always non-negative because $e^0 = 1$, ensuring that the trace term is always at least 1 resulting in $\ln(1) = 0$.

We now add the non-negative terms $(1), (2), (3), (4), (5)$ and observe that the inequality:
$$D\big(\rho_{\mathbf{A+B}}\|\rho_{\mathbf{B}}\big) \;+\; D\big(\rho_{\mathbf{A+B}}\|\rho_{\mathbf{A}}\big) \;+\; \ln Z_{\mathbf{C}} \;+\; \beta\,\mathrm{Tr}[\mathbf{A}\,\rho_{\mathbf{A}}] \;+\; \beta\,\mathrm{Tr}[\mathbf{B}\,\rho_{\mathbf{B}}] \;\geq 0$$

We then exploit the fact that for a Gibbs-like state the Von Neumann entropy is given by $S(\mathbf{A+B}) = \beta\,\mathrm{Tr}[(\mathbf{A+B})\rho_{\mathbf{A+B}}] + \ln Z_{\mathbf{A+B}}$. After expanding out the KL divergences, recalling that $\mathbf{C} = \mathbf{A+B}$, and re-arranging, we get:
$$-2\,S(\mathbf{A+B}) + \beta\,\mathrm{Tr}[(\mathbf{A+B})\rho_{\mathbf{A+B}}] + \ln Z_{\mathbf{A+B}} + S(\mathbf{A}) + S(\mathbf{B}) \;\geq 0.$$

This, after some basic algebraic manipulation, allows us to conclude

$$S(\mathbf{A} + \mathbf{B}) \leq S(\mathbf{A}) + S(\mathbf{B}).$$

Thus $\boxed{S(\mathbf{A} + \mathbf{B}) \leq S(\mathbf{A}) + S(\mathbf{B})}$.

Finally, recall $\mathbf{A} = \widetilde{\mathbf{C}}^1$ and $\mathbf{B} = \widetilde{\mathbf{C}}^2$, so

$$S(\widetilde{\mathbf{C}}^1 + \widetilde{\mathbf{C}}^2) \leq S(\widetilde{\mathbf{C}}^1) + S(\widetilde{\mathbf{C}}^2).$$

But each $\widetilde{\mathbf{C}}^j$ has the same entropy as $\mathbf{C}^j$, and $\widetilde{\mathbf{C}}^1 + \widetilde{\mathbf{C}}^2$ has the same entropy as $\mathbf{C}^1 + \mathbf{C}^2$. So

$$S\big(\mathbf{C}^1 + \mathbf{C}^2\big) = S(\widetilde{\mathbf{C}}^1 + \widetilde{\mathbf{C}}^2) \leq S(\widetilde{\mathbf{C}}^1) + S(\widetilde{\mathbf{C}}^2) = S(\mathbf{C}^1) + S(\mathbf{C}^2).$$

This completes the base case $n = 2$ in all detail.

**Inductive Step.** Suppose for some $k \geq 2$, $S\big(\sum_{j=1}^{k} \mathbf{C}^j\big) \leq \sum_{j=1}^{k} S(\mathbf{C}^j)$. We show it for $k + 1$:

$$\sum_{j=1}^{k+1} \mathbf{C}^j = \Big(\sum_{j=1}^{k} \mathbf{C}^j\Big) + \mathbf{C}^{k+1}.$$

By the $n = 2$ sub-additivity (applying it to $\sum_{j=1}^{k} \mathbf{C}^j$ and $\mathbf{C}^{k+1}$),

$$S\Big(\sum_{j=1}^{k} \mathbf{C}^j + \mathbf{C}^{k+1}\Big) \leq S\Big(\sum_{j=1}^{k} \mathbf{C}^j\Big) + S(\mathbf{C}^{k+1}).$$

But by induction hypothesis, $S\big(\sum_{j=1}^{k} \mathbf{C}^j\big) \leq \sum_{j=1}^{k} S(\mathbf{C}^j)$. Therefore

$$S\Big(\sum_{j=1}^{k+1} \mathbf{C}^j\Big) \leq \sum_{j=1}^{k} S(\mathbf{C}^j) + S(\mathbf{C}^{k+1}) = \sum_{j=1}^{k+1} S(\mathbf{C}^j).$$

Hence by induction, for any $n$,

$$\boxed{S\Big(\sum_{j=1}^{n} \mathbf{C}^j\Big) \leq \sum_{j=1}^{n} S(\mathbf{C}^j).}$$

This completes the proof. $\qquad\square$

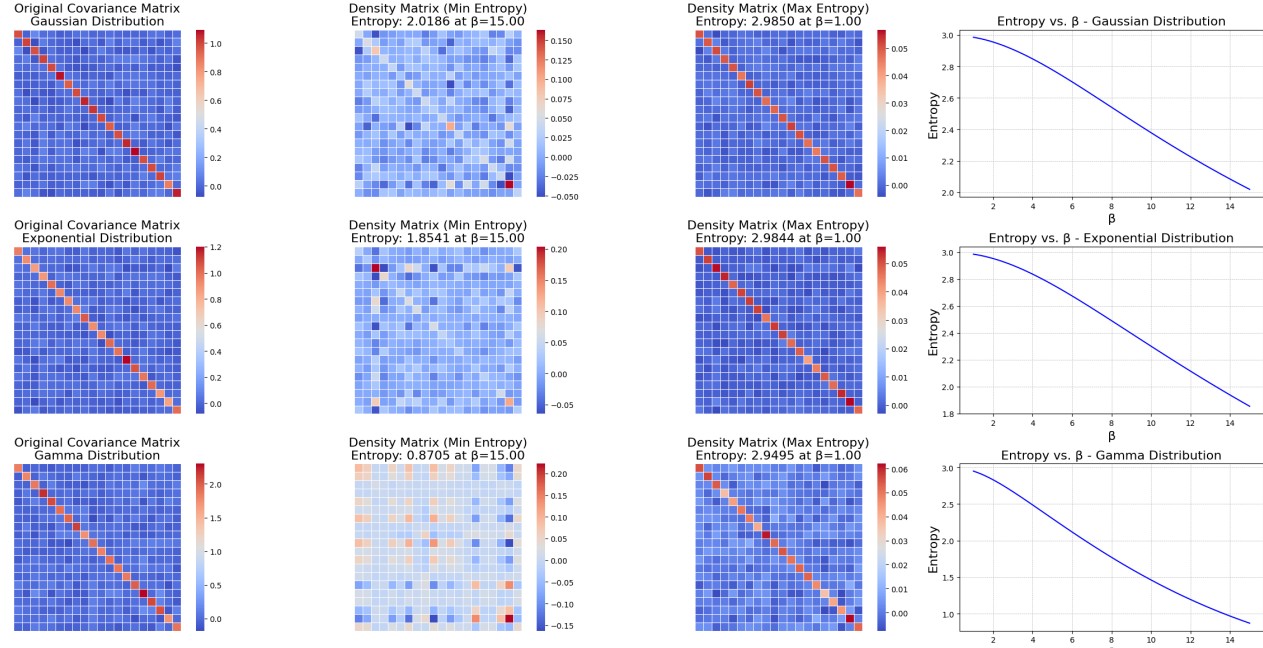

Figure 9: We sample covariance matrices from Gaussian,Exponential and Gamma distributions and observe the behaviour for different values of beta. Higher values of $\beta$ relate to increased diffusion across the covariance matrix and thus reduced regularity and thus entropy, smaller values have higher entropy. We can thus conclude that CDNN's in a higher entropy state are more stable, corresponding to traditional thermodynamical principles.

### A.8 Multi–scale Von–Neumann Entropy for *Singular* Gaussians

Let $\mathbf{X} = (X, Y, Z)^\top \sim \mathcal{N}(\mathbf{0}, \Sigma)$ with $\mathrm{rank}(\Sigma) = r < p = 3$. The classical differential entropy

$$h(\mathbf{X}) \;=\; \frac{p}{2}\Big[1 + \log(2\pi)\Big] \;+\; \frac{1}{2}\log\det\Sigma \tag{10}$$

is defined w.r.t. Lebesgue measure on $\mathbb{R}^3$. If $\det\Sigma = 0$ the right term diverges to $-\infty$, so every closed-form Gaussian entropy estimator *explodes*. We show that the multi-scale Von–Neumann entropy (VNE) overcomes this obstruction and can distinguish singular covariances that differ only by a global scale.

**Proposition 6** (Density matrix "lifts" singular covariances). *Let $\Sigma \succeq 0$ be a $p \times p$ covariance matrix with $\mathrm{rank}\,\Sigma = r < p$ (hence $\det\Sigma = 0$). For any finite $\beta \neq 0$ define the density matrix*

$$\rho_\beta(\Sigma) \;=\; \frac{\exp(-\beta\Sigma)}{\mathrm{tr}\exp(-\beta\Sigma)}.$$

*Then*

$$\rho_\beta(\Sigma) \succ 0 \quad and \quad \det\rho_\beta(\Sigma) \;>\; 0.$$

*Proof.* Diagonalise $\Sigma = Q\,\mathrm{diag}(\lambda_1, \ldots, \lambda_r, \underbrace{0, \ldots, 0}_{p-r})Q^\top$ with $Q$ orthogonal and $\lambda_i > 0$ for $i \leq r$. Matrix–exponential in the same basis is

$$\exp(-\beta\Sigma) = Q\,\mathrm{diag}\big(e^{-\beta\lambda_1}, \ldots, e^{-\beta\lambda_r}, \underbrace{1, \ldots, 1}_{p-r}\big)Q^\top,$$

whose eigen-values are $e^{-\beta\lambda_1}, \ldots, e^{-\beta\lambda_r}$ and 1 (repeated $p - r$ times). All of them are *strictly* positive, hence $\exp(-\beta\Sigma) \succ 0$ and

$$\det\big[\exp(-\beta\Sigma)\big] \;=\; e^{-\beta\sum_{i=1}^{p}\lambda_i} \;>\; 0.$$

Dividing by the positive scalar $\operatorname{tr} \exp(-\beta\Sigma) = \sum_{i=1}^{p} e^{-\beta\lambda_i}$ preserves positive–definiteness and scales every eigen-value by the same constant $c^{-1}$. Therefore

$$\rho_\beta(\Sigma) \succ 0, \qquad \det\rho_\beta(\Sigma) = \frac{e^{-\beta\sum_i \lambda_i}}{\left(\sum_i e^{-\beta\lambda_i}\right)^p} > 0. \qquad \square$$

Proposition 6 guarantees that the Von–Neumann entropy $S_\beta(\Sigma)$ is finite even when $\Sigma$ is rank–deficient, circumventing the divergence of the classical $\frac{1}{2}\log\det\Sigma$ term.

Take two different rank-2 covariances (both det $= 0$):

$$\Sigma_1 = \begin{pmatrix} 2 & 0 & 0 \\ 0 & 0 & 0 \\ 0 & 0 & 0 \end{pmatrix}, \qquad \Sigma_2 = \begin{pmatrix} 1 & 0 & 0 \\ 0 & 1 & 0 \\ 0 & 0 & 0 \end{pmatrix}.$$

For $\beta = 1$ their Von–Neumann entropies are

$$S_1(\Sigma_1) = 1.28 \text{ bits}, \qquad S_1(\Sigma_2) = 1.41 \text{ bits}.$$

Both matrices are singular, hence the log–det term diverges, but the VNE still produces finite, *different* values—lower for the "collapsed" $\Sigma_1$, higher for the more evenly spread $\Sigma_2$. Thus VNE distinguishes degrees of randomness even among rank-deficient covariances where classical formulas fail.

Now consider a commonly used naive estimator of the entropy of the covariance matrix.

$$S_{\text{naive}}(\Sigma) \;=\; -\sum_{i=1}^{p} \pi_i \log_2 \pi_i, \qquad \pi_i \;=\; \frac{\lambda_i}{\operatorname{tr}\Sigma}, \tag{11}$$

We begin by noting the following observation

**Proposition 7** (Scale blindness of the trace–normalised surrogate)**.** *Let* $\Sigma_1 \succeq 0$ *have rank* $r < p$ *and* $\Sigma_2 = \alpha\Sigma_1$ *with* $\alpha > 0$*. Then* $S_{\text{naive}}(\Sigma_2) = S_{\text{naive}}(\Sigma_1)$*.*

*Proof.* Scaling each non-zero eigen-value by $\alpha$ multiplies $\operatorname{Tr}\Sigma$ by the same factor, leaving every ratio $\pi_i = \lambda_i/\operatorname{Tr}\Sigma$ unchanged; hence equation 11 is invariant. $\qquad\square$

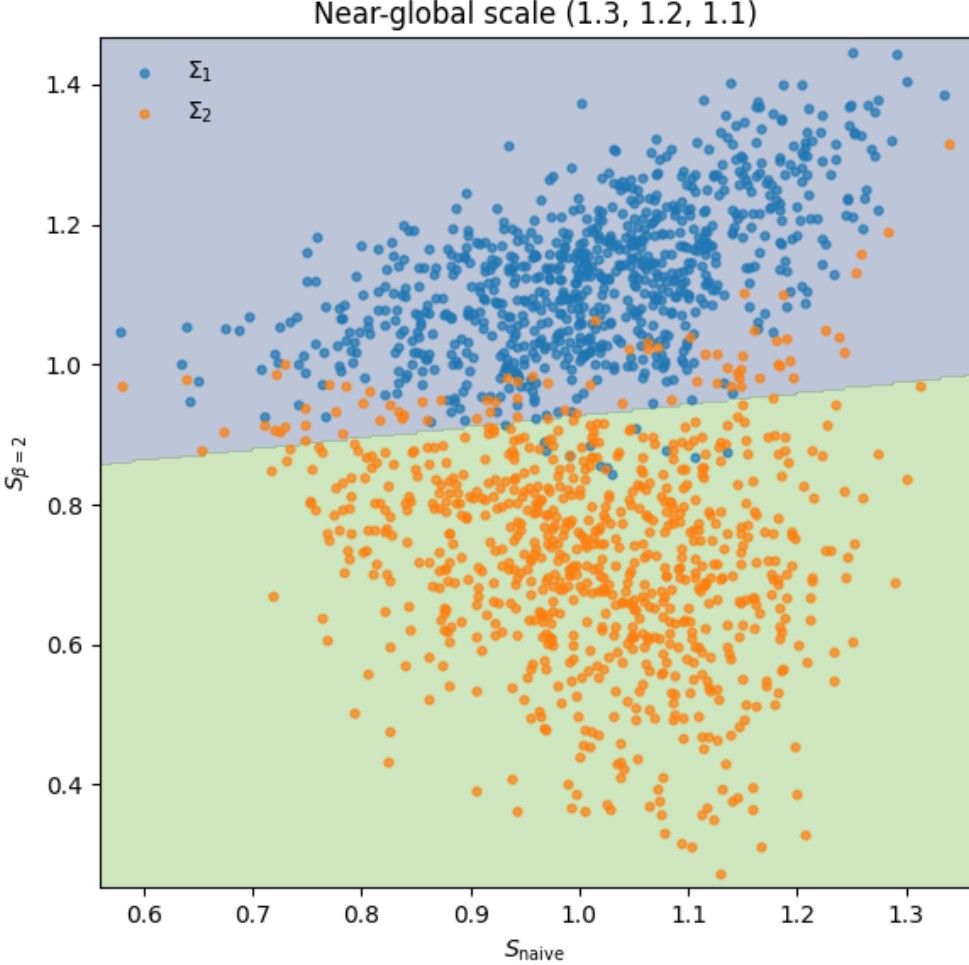

Figure 10: Scatter of naïve entropy $S_{\text{naive}}$ versus von Neumann entropy $S_{\beta=2}$ for two covariance regimes $\Sigma_1$ (blue) and $\Sigma_2$ (orange) at near–global scale (eigenvalues multiplied by $(1.3, 1.2, 1.1)$).

At the extreme global-scale limit (e.g. uniform multiplication of all eigenvalues by a large factor), the naïve entropy remains exactly constant and is therefore blind to any change in overall variance magnitude. Even when the scale change is mild,as in Figure 10, where the spectrum shifts by a element wise multiplication of $(1.3, 1.2, 1.1)$,both regimes overlap almost perfectly along the $S_{\text{naive}}$ axis, indicating that normalized-spectrum methods cannot distinguish them.

By contrast, von Neumann entropy at $\beta = 2$, remains sensitive to absolute eigenvalue differences even when ratios are nearly equal. In the same near–global-scale setting, $S_{\beta=2}$ cleanly separates the two clusters into distinct vertical bands, capturing both the slight change in global variance and the subtle reshaping of the spectrum.

Quantitatively, a simple LDA classifier thresholded on $S_{\text{naive}}$ yields an AUC of only 0.4992, essentially chance, whereas using $S_{\beta=2}$ achieves an AUC of 0.9869, demonstrating that von Neumann entropy vastly outperforms the naïve measure at detecting even mild global-scale changes. In cases such as spike detection in neurological signals, the naive entropy estimator may miss out crucial near global changes that are common neural responses.

Of course, there may be cases where invariance to scale may be desirable, thus CVNE acts as a *complement* to existing entropic measures, rather than a replacement.

## A.9   Reconstructing the True Covariance Matrix

Reconstructing the original covariance matrix involves formulating and solving an optimization problem that connects the spectral Properties of the covariance matrix to a density-like representation.

Begin with a symmetric positive-definite covariance matrix $\mathbf{C}$ that can be decomposed as

$$\mathbf{C} \;=\; \mathbf{U}\,\mathbf{\Lambda}\,\mathbf{U}^T,$$

where $\mathbf{U}$ is orthogonal and $\mathbf{\Lambda} = \mathrm{diag}(\lambda_1,\ldots,\lambda_n)$ contains the eigenvalues. The *target distribution* is obtained by normalising the eigenvalues,

$$p_i \;=\; \frac{\lambda_i}{\sum_j \lambda_j}, \qquad i = 1,\ldots,n.$$

At the same time we define the *covariance–density distribution* parameterised by an inverse temperature $\beta \in \mathbb{R}$,

$$q_{\beta,i} \;=\; \frac{\exp(-\beta\lambda_i)}{\sum_j \exp(-\beta\lambda_j)}.$$

**Definition 8** (Moment objective).  *Let*

$$f(\beta) \;=\; \beta \sum_{i=1}^{n} p_i\lambda_i \;+\; \log\!\Big(\sum_{j=1}^{n} e^{-\beta\lambda_j}\Big),$$

*obtained by dropping the constant term $\sum_i p_i \log p_i$ from the Kullback–Leibler divergence $D_{\mathrm{KL}}(\mathbf{p}\,\|\,\mathbf{q}_\beta)$.*

**Proposition 8** (Strict convexity).  *The function $f\colon \mathbb{R} \to \mathbb{R}$ of Definition 8 is twice continuously differentiable and* strictly convex. *Explicitly,*

$$f'(\beta) = \sum_{i=1}^{n} p_i\lambda_i \;-\; \frac{\sum_{j=1}^{n} \lambda_j e^{-\beta\lambda_j}}{\sum_{j=1}^{n} e^{-\beta\lambda_j}}, \qquad f''(\beta) = \mathrm{Var}_{\mathbf{q}_\beta}[\lambda] \;>\; 0,$$

*unless all eigenvalues are identical (a degenerate case).*

*Proof.* Let $Z(\beta) = \sum_j e^{-\beta\lambda_j}$. Then $f'(\beta) = \sum_i p_i\lambda_i - \dfrac{Z'(\beta)}{Z(\beta)}$. Because $Z'(\beta) = -\sum_j \lambda_j e^{-\beta\lambda_j}$,

$$f'(\beta) = \sum_i p_i\lambda_i - \frac{\sum_j \lambda_j e^{-\beta\lambda_j}}{\sum_j e^{-\beta\lambda_j}} = \sum_i p_i\lambda_i - \mathbb{E}_{\mathbf{q}_\beta}[\lambda].$$

Differentiating once more and applying the quotient rule gives $f''(\beta) = \mathbb{E}_{\mathbf{q}_\beta}[\lambda^2] - \mathbb{E}_{\mathbf{q}_\beta}[\lambda]^2 = \mathrm{Var}_{\mathbf{q}_\beta}[\lambda]$. Because the eigenvalues are not all equal, this variance is strictly positive, so $f$ is strictly convex and $C^2$. □

**Theorem 9** (Existence and uniqueness of the global minimiser).  *There exists a* unique *value $\beta^\star \in \mathbb{R}$ satisfying*

$$f'(\beta^\star) = 0 \quad \Longleftrightarrow \quad \sum_{i=1}^{n} p_i\lambda_i = \frac{\sum_{j=1}^{n} \lambda_j e^{-\beta^\star\lambda_j}}{\sum_{j=1}^{n} e^{-\beta^\star\lambda_j}} = \mathbb{E}_{\mathbf{q}_{\beta^\star}}[\lambda].$$

*This $\beta^\star$ is the **global minimiser** of $f(\beta)$ and therefore also minimises $D_{\mathrm{KL}}(\mathbf{p}\,\|\,\mathbf{q}_\beta)$.*

*Proof.* By Theorem 8, $f$ is strictly convex, hence possesses at most one stationary point. Recall that $f'(\beta) = \bar{\lambda}_p - \mathbb{E}_{\mathbf{q}_\beta}[\lambda]$, where $\bar{\lambda}_p = \sum_i p_i\lambda_i$. Because the $p_i$ are strictly positive and sum to one, $\bar{\lambda}_p$ is a strict convex combination of the eigenvalues; when these are not all equal this gives $\lambda_{\min} < \bar{\lambda}_p < \lambda_{\max}$. As $\beta \to -\infty$ the measure $\mathbf{q}_\beta$ concentrates on $\lambda_{\max}$, so $\lim_{\beta\to-\infty} f'(\beta) = \bar{\lambda}_p - \lambda_{\max} < 0$; as $\beta \to +\infty$ it concentrates on $\lambda_{\min}$, so $\lim_{\beta\to+\infty} f'(\beta) = \bar{\lambda}_p - \lambda_{\min} > 0$. Since $f'$ is continuous and changes sign, the intermediate value theorem guarantees the existence of a $\beta^\star$ with $f'(\beta^\star) = 0$. Strict convexity then makes this point unique and global. □

With $\beta^\star$ obtained (*e.g.* numerically via Gradient Descent), the optimal covariance-density distribution is

$$q_{\beta^\star,i} = \frac{\exp(-\beta^\star \lambda_i)}{\sum_j \exp(-\beta^\star \lambda_j)},$$

and the associated *density matrix* is reconstructed as

$$\mathbf{M} = \frac{\exp(-\beta^\star \mathbf{C})}{\mathrm{tr}\big(\exp(-\beta^\star \mathbf{C})\big)}.$$

We generate a covariance matrix from a standard Gaussian matrix and use a BFGS routine to recover $\beta^\star$. Figures 11 and 12 display the original matrix, its reconstruction, and the alignment of the eigenvalue distribution. Consequently, a *Covariance Density Network* equipped with the multi-scale filter bank $\{\beta^\star\}$ is generally capable of matching (and often surpassing) the representational power of a classic Covariance Neural Network.

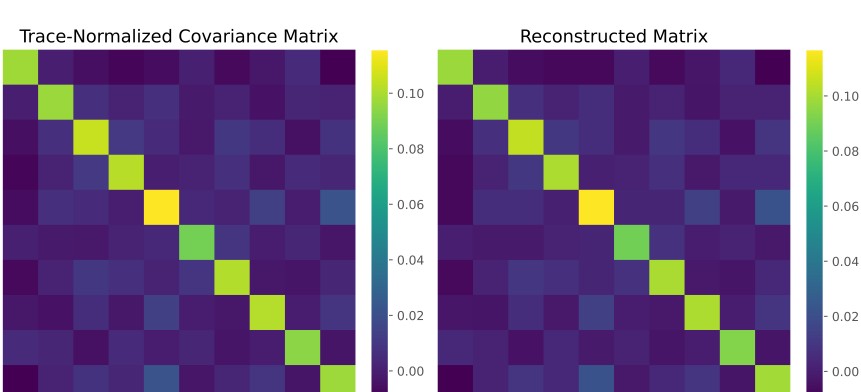

Figure 11: Original and reconstructed covariance matrices.

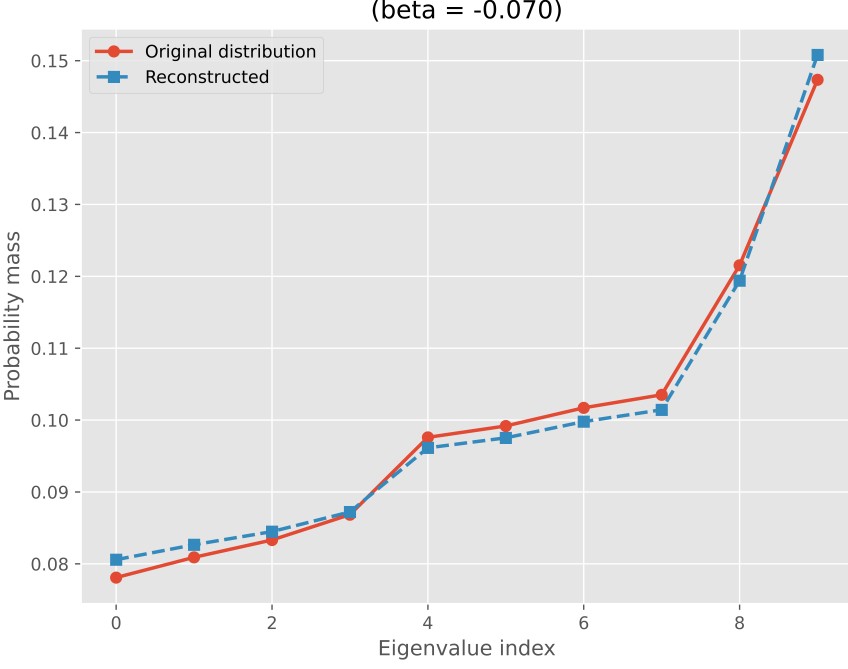

Figure 12: Eigenvalues of reconstructed and original covariance matrix.

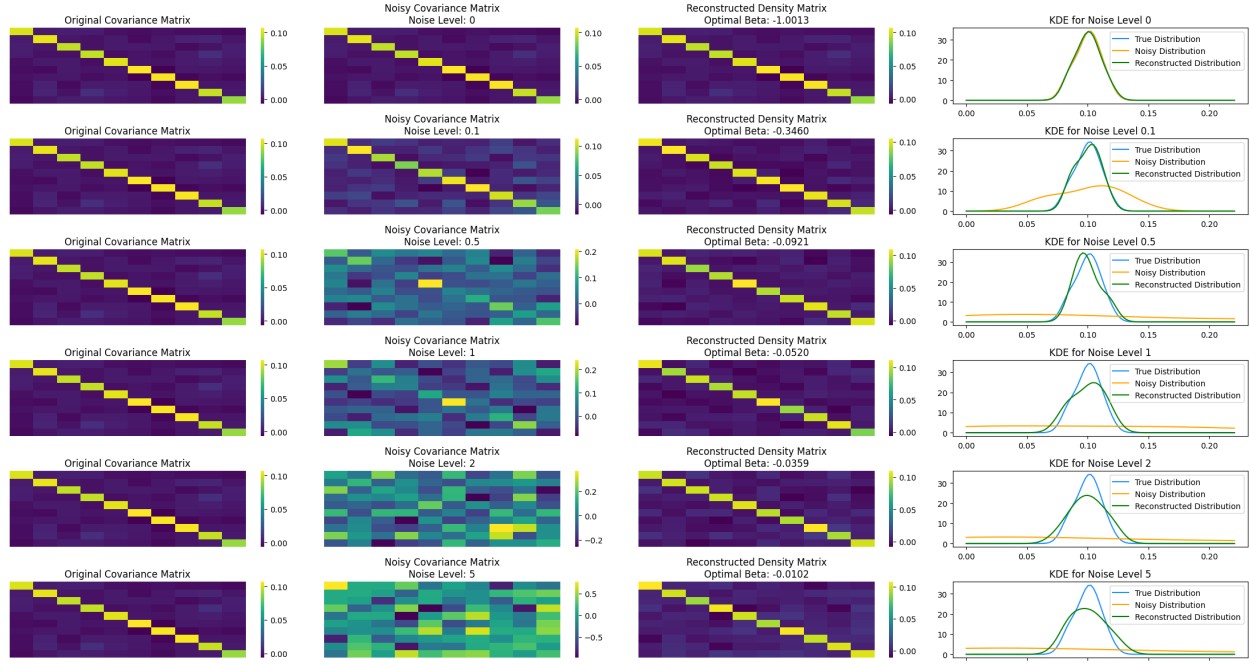

Figure 13: Synthetic example: Gaussian data generate $\mathbf{C}_{\text{true}}$; additive noise yields $\mathbf{C}_{\text{noisy}}$. Matching the noisy and true eigenspectra via $\beta$ produces a density matrix that closely approximates $\mathbf{C}_{\text{true}}$. See the connection to **Minimum Probability Flow Learning** (Sohl-Dickstein et al., 2011).

We can also match *realistic* distributions by optimizing the beta parameter. For example, we can construct a Covariance Density Matrix with a noisy estimate of the true Covariance and optimize beta to match the noisy estimate to the true one (See Figure 10).

While Figure 13 shows the case when we recreate a covariance matrix from *standard* normal distrbution. We can also consider more challenging cases i.e covariance matrices from auto-regressive processes with non-trivial off-diagonal elements. A more intuitive approach could also be to reconstruct the precision matrix as this would lie in the positive $\beta$ domain.

Figure 14 shows the eigenvalue distribution reconstruction of the true precision and covariance matrix from highly correlated auto-regressive data. We note that while the actual entry by entry reconstruction of the covariance matrix was not ideal but the eigenvalue distribution was well recovered. We encourage future work in this domain.

**Empirical Stability Analysis**

### A.9.1 Stability Analysis of the Operator Norm

The Operator norm difference between original and perturbed matrices is used to evaluate stability.

In particular we measure:
$$\|\Delta\| = \|\mathbf{C}_{\text{original}} - \mathbf{C}_{\text{perturbed}}\|,$$
and similarly for the density matrices:

$$\|\Delta\mathbf{P}\| = \|\mathbf{P}_{\text{original}} - \mathbf{P}_{\text{perturbed}}\|.$$

Where $\mathbf{P}_{\text{perturbed}}$ is a perturbation only in the covariance matrix used to compute $\mathbf{P}$.

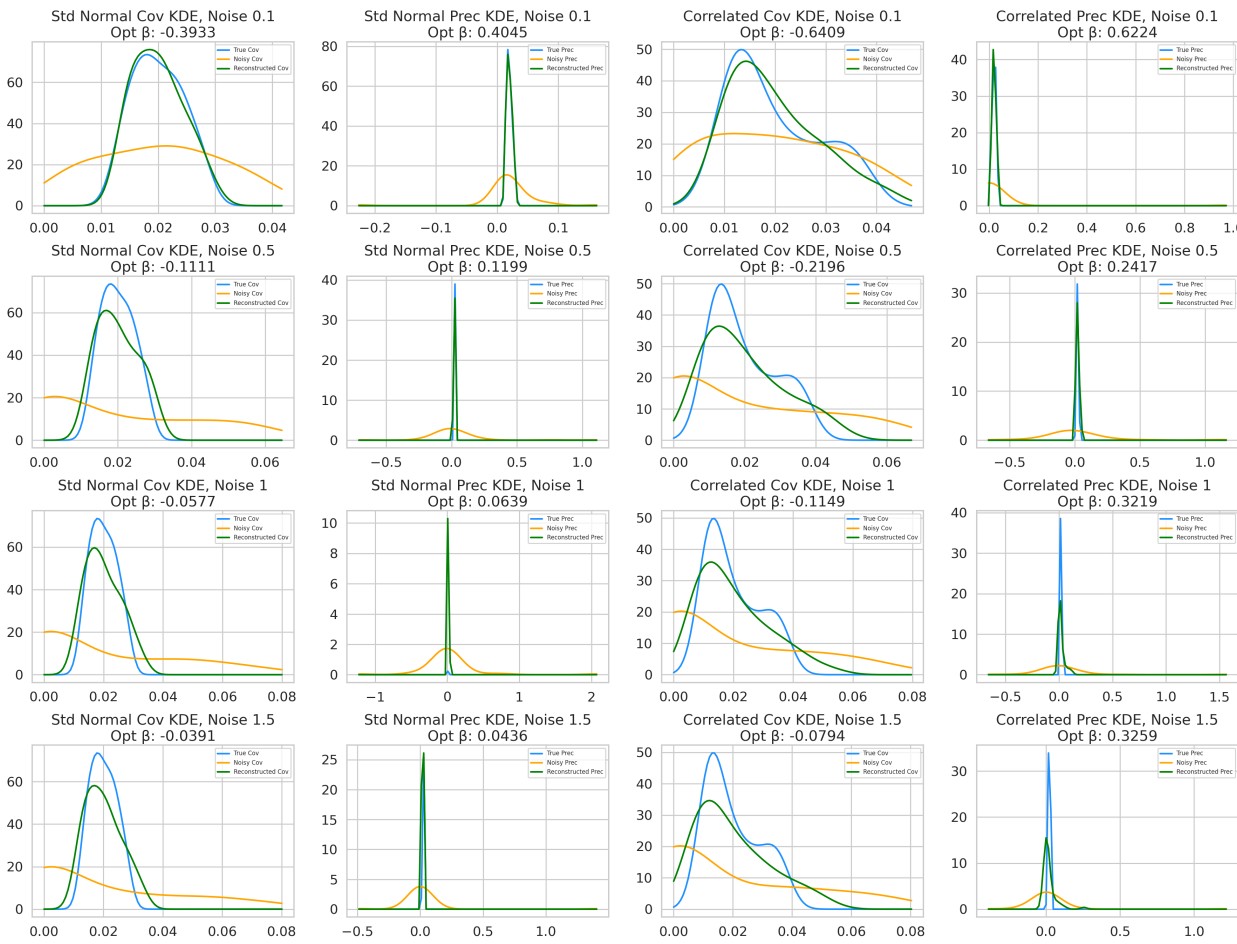

Figure 14: Reconstructing the eigenvalue distribution of a (Standard Normal vs highly correlated) Covariance and Precision matrix from a noise perturbed estimate using our moment matching routine.

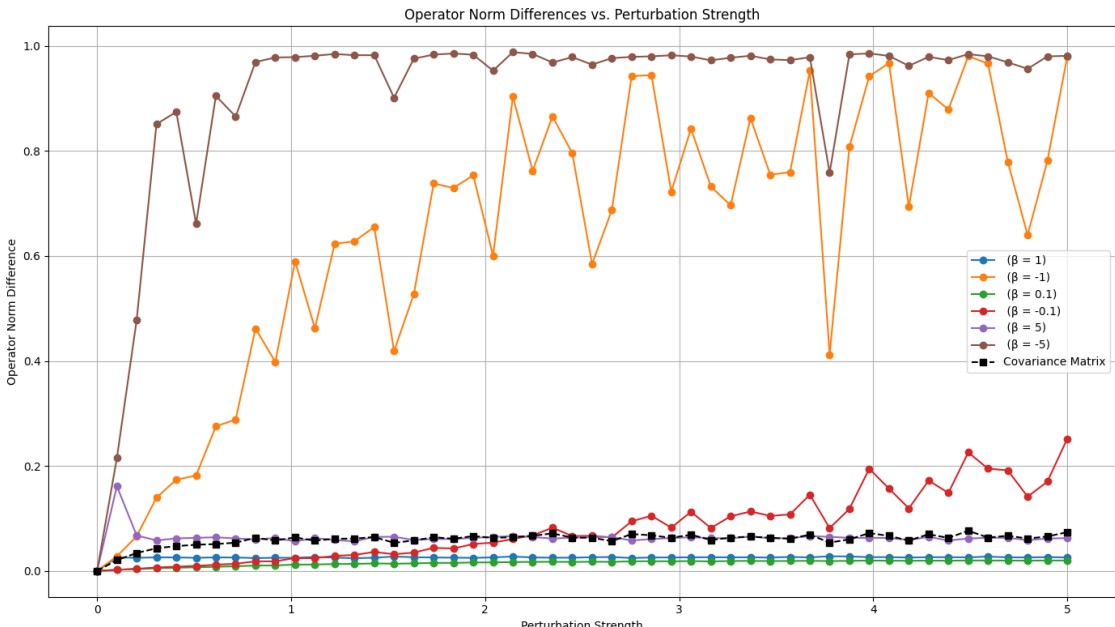

Figure 15: Stability of Covariance Density matrices to perturbations in the sample covariance matrix under the Operator norm.

We generate covariance matrices from Gaussian data and add noise perturbations at different levels. Observe that we pay large penalties for negative $\beta$; however, this tends to decrease as $\beta$ tends to 0. Positive values of $\beta$ show enhanced stability compared to the trace-normalized covariance matrix, although as $\beta$ increases, the stability decreases. This decrease is much less pronounced than in the negative $\beta$ scenario. Figure 10 thus corresponds well with Lemma 1.

### A.9.2 Stability of CDNN in Synthetic Regression problem

After a Taylor Expansion $\mathbf{P}$ can be expressed as:

$$\mathbf{P} = \frac{\mathbf{I} - \beta\mathbf{C} + \frac{\beta^2\mathbf{C}^2}{2!} - \frac{\beta^3\mathbf{C}^3}{3!} + \cdots}{Z}. \tag{12}$$

As we purely want to observe the effect of $\mathbf{C}$ and its transformations on the stability to the sample size of the covariance matrix, we update $\mathbf{x}$ as:

$$\mathbf{x}_{\text{shifted}} = \mathbf{Px} - \frac{\mathbf{x}}{Z}, \tag{13}$$

where $\mathbf{P}$ applies the density matrix $\mathbf{P}$ to the input $\mathbf{x}$.

As in Sihag et al. we compare the stability of CDNNs relative to perturbations in the sample covariance matrix, i.e. we vary the number of samples used for the construction of the covariance matrix. We replicate the exact conditions in Sihag et al. and compare our approach with VNNs, Linear regression with PCA components, and PCA with a Radial Basis Function (RBF) Kernel on random linear regression problems using the routine `sklearn.datasets.make_regression` in Python, which lets us specify various parameters. We generate two cases, one with no external noise and one with a noise level of 5 (a parameter we can tune directly in the Python dataset generation).

Figure 6 shows the regression performance under no noise. We see that at smaller values of $\beta$ we maintain almost perfect stability; however, the MAE performance is weaker than VNNs. We conjecture that in the Friedmann regression problem (Breiman, 1996), eigenmodes corresponding to lower-variance principal components are the most discriminating, and since VNNs inherently discriminate these components better, they achieve strong performance. At small values of $\beta$ these low-variance components are not as discriminable

and thus the performance suffers, but as $\beta$ increases to larger values we can clearly see that as more low-variance components are shifted to the discriminable eigenspace the performance improves, and at $\beta = 15$ we see improved performance compared to VNNs.

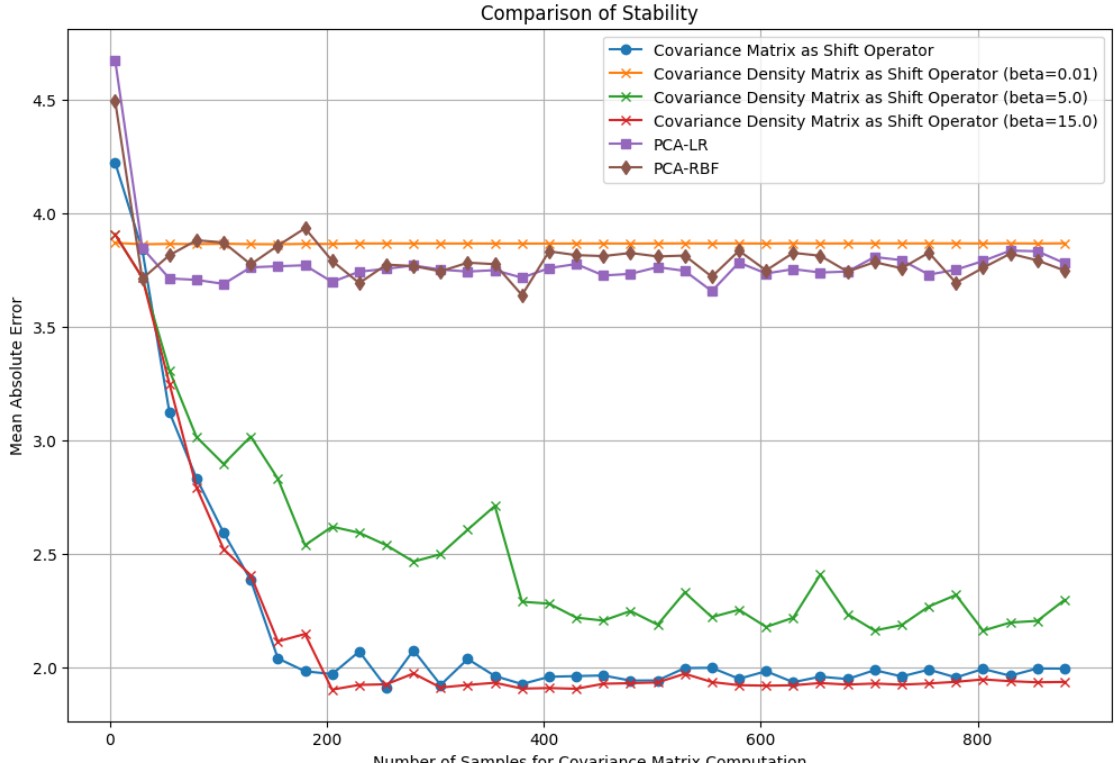

Figure 16: Regression performance under noise-free conditions.

Under noisy conditions, regression models based on density matrices ($\boldsymbol{\rho}$) showed superior stability compared to covariance neural networks.

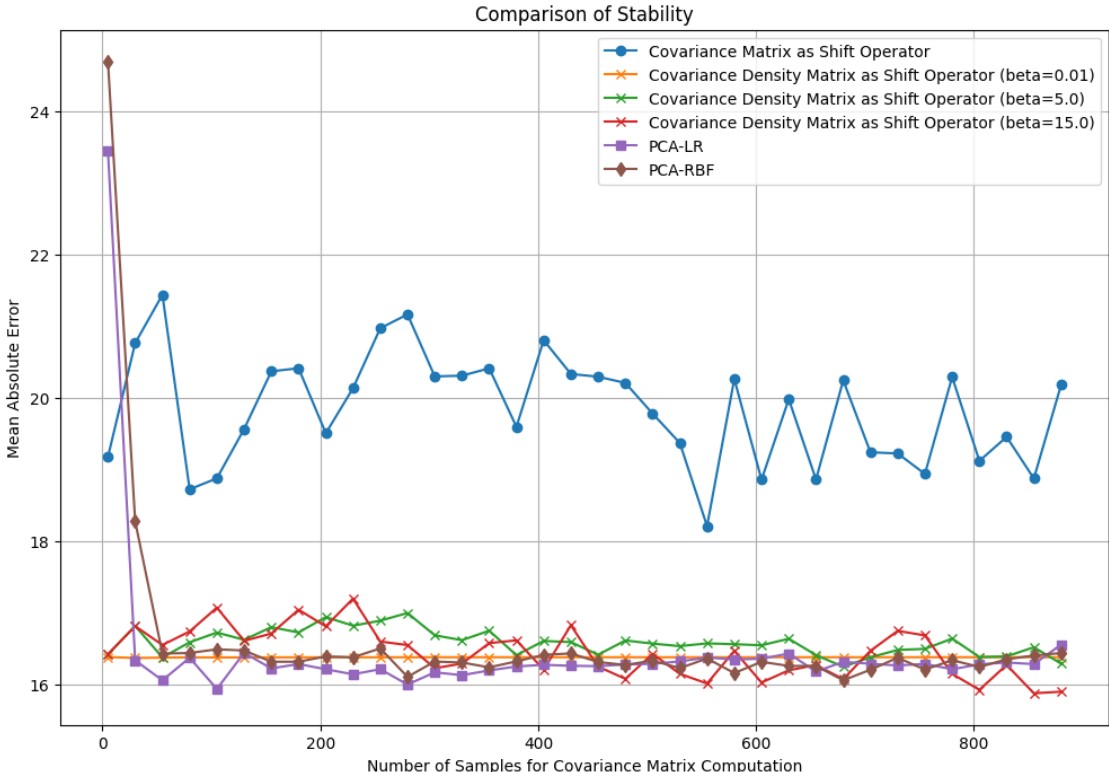

Figure 17: Regression performance under noisy conditions.

Figure 7 repeats the experiment with the same conditions but the noise level is increased to 5. We can see that VNNs now suffer a significant drop in performance. This can be attributed to the noise being mostly concentrated in the low-variance components (i.e. the eigenspace that VNNs are best able to discriminate), significantly reducing performance. CDNNs discriminate best in the high-variance eigenspace, so the noise is less likely to affect performance. Thus, even at small $\beta$ we see stable and stronger performance than VNNs. PCA also operates in this high-variance space and thus does not suffer from noise as much. This suggests that CDNNs inherently exhibit a greater robustness to external noise, regardless of the value of *positive* $\beta$.

