# OpenReview forum: "Covariance Density Neural Networks"
_TMLR — Accepted by TMLR_

### Review · Reviewer_HrxG · 2025-12-19

**Summary Of Contributions:**

Standard GNNs require a pre-specified graph to define connections between nodes; in many cases, this graph is unknown and difficult to design without strong prior knowledge. This paper proposes a covariance-driven alternative by constructing the graph operator directly from data. Its core contribution is to replace the covariance shift operator with a covariance density operator and use (optionally multi-scale) graph filtering to mix node features. The authors provide grounded theoretical development and show improvements over baselines on financial forecasting and EEG classification tasks.

**Audience:**

Yes

**Audience Explanation:**

This works provides a plug-and-play module for constructing graph operators directly from data,  which is a timely and useful contribution for the community.

**Claims And Evidence:**

Yes

**Claims Explanation:**

Claims are well supported by theoretical development and empirical results. The authors are also honest about the limitations and explicitly acknowledged that current work is limited to linear dependency and stationary data.

**Requested Changes:**

**Q1** I feel confused about how to calculate the covariance matrix $C$ for data from different domains (e.g time-histories data has a time dimension along with the feature dimension, while other type of data may only have feature dimension) . What constitutes one "observation"? (timepoints or trials for EEG data? )

**Q2** The term “quasi-Hamiltonian” is best understood as a spectral analogy rather than a literal physical construct. I recommend adding an explicit clarifying sentence in Section 2 stating that the terminology is metaphorical and is intended to highlight the role of (C) in shaping the spectrum, not to claim a strict physical interpretation.

**Q3**  typos  page 2. . “how it alllows us to control” ->  allows

page 8 “ added covariance information is indeed informative and that that that”, remove extra “that”

Proof of Theorem 1 : “respect to the denisty transformation” -> density

**Q4** citation style, please use standard TMLR citation format (authors, year)

---

> ### Author Response · Authors · 2026-02-14
>
> We sincerely thank Reviewer HrxG for the positive and constructive evaluation. We address each point below.
>
> Q1: How to calculate the covariance matrix for different domains?
> Thank you for raising this important clarification. We have added a new paragraph at the beginning of Section 4 (Experimental Results) that explicitly describes the covariance matrix construction for each domain. In brief:
>
> Financial data: Each asset/feature constitutes a random variable (node), and each time step within a sliding window constitutes one observation. The sample covariance matrix is computed once over the full training set and captures cross-asset correlations.
>
> EEG data: Each EEG channel is a random variable (node), and each time step in a trial constitutes one observation. The covariance matrix is computed once over all training subjects' trials timesteps, repeated for all participants and averaged over, producing a global cross-channel covariance that captures shared spatial patterns across individuals.
>
> "Quasi-Hamiltonian" is metaphorical.
> We agree and have added an additional clarifying footnote in Section 2: "We emphasise that the term 'quasi-Hamiltonian' is used as a spectral analogy ,highlighting the role of $\mathbf{C}$ in shaping the system's spectral modes and governing signal diffusion , rather than implying a strict quantum-mechanical interpretation."
>
> Q3: Typos.
> All three typos have been corrected:
>
> Page 2: "alllows" → "allows"
> Page 8: removed extra "that"
> Proof of Theorem 1: "denisty" → "density"
>
> Q4: Citation style.
> We have converted all citations to the standard TMLR (authors, year) format throughout the paper.

---

### Review · Reviewer_wk2E · 2026-01-07

**Summary Of Contributions:**

Summary and Overall Assessment

The paper introduces Covariance Density Neural Networks (CDNNs), proposing to construct a density matrix from a covariance matrix and use it as a graph operator in a GNN-like architecture. The authors present some theoretical analyses and empirical studies in financial forecasting and EEG classification.

My evaluation is largely negative, which could be a result of my unfamiliarity of the field. That being said, the paper lacks a clear motivation and fails to articulate what problem it actually aims to solve. For a reader not already familiar with the graph-signal-processing lineage of GNNs, it is extremely unclear what CDNNs are designed for, why this formulation is needed, and what concrete advantage it offers over prior covariance-based or diffusion-kernel methods. The exposition assumes substantial background knowledge without providing a proper introduction or positioning within the existing literature.

**Audience:**

Yes

**Audience Explanation:**

The paper is relevant to some audience obviously. Still, it should clarify the background and relevant literatures substantially to attract people who was not familiar with graph-signal-processing and GNNs.

**Claims And Evidence:**

No

**Claims Explanation:**

Main Concerns

1. Poorly defined problem and insufficient motivation.
The introduction does not identify a specific gap or limitation in existing GNN or covariance-based frameworks that CDNN meaningfully resolves. The description of the model reads as a reparameterization of the CoVariance Neural Network (VNN) with a temperature-like parameter β, but the reader is left guessing what benefit this confers in practice.

2. Lack of clarity and accessibility.
The background material is underdeveloped. Without familiarity with prior work on VNNs or graph diffusion kernels, the technical construction appears unmoored—there is no intuitive explanation of when one would want to use this operator or what kind of data it is intended for. A substantially expanded literature review and clearer motivation are needed to make the contribution intelligible to a general ML audience.

3. Triviality of theoretical analysis.
Theoretical results (equivariance, Lipschitz and stability bounds) appear to be routine adaptations of standard GNN analyses rather than novel insights. They mainly confirm expected behavior of smooth spectral filters and do not establish new principles.

4. Empirical results difficult to interpret.
The experiments show some improvements, but the paper does not convincingly link those gains to the proposed method. As I am not a specialist in GNNs, I cannot judge how compelling the empirical evidence is within that subfield. I am willing to defend my assessment of weak motivation and conceptual triviality, but I acknowledge that I might be missing the paper’s intended main point.

Recommendation

The work would benefit from a much clearer articulation of the target problem, a grounded comparison to existing covariance- and diffusion-based graph methods, and a deeper explanation of why the proposed construction is necessary. In its current form, the contribution and novelty remain ambiguous, and the theoretical analysis adds little substance beyond standard results.

Again, although I'm willing to defend my assesment, I have to admit the paper falls beyond my scope of expertise.

**Requested Changes:**

Please see my comments above.

---

> ### Author Response · Authors · 2026-02-14
>
> We thank Reviewer wk2E for their time and response. We appreciate the acknowledgment that this paper may fall outside the reviewer's primary expertise, and we address each concern in detail.
>
> 1. "Poorly defined problem and insufficient motivation":
>
> We respectfully disagree that the problem is poorly defined, but we acknowledge that the original manuscript did not lead with the motivation clearly enough. We have rewritten the Introduction (Section 1) to lead with the concrete problem:
>
> To summarise our new introduction we state:
>
> "In many domains ,neuroscience, finance, genomics , we observe multivariate signals but have no known graph topology. Standard GNNs require a pre-specified graph. CoVariance Neural Networks (VNNs) address this by using the sample covariance matrix as a graph shift operator. However, VNNs have two key limitations: (1) they cannot control which spectral components are discriminable versus stable, and (2) they are sensitive to noise concentrated in low-variance eigenmodes. CDNNs resolve both issues. "
>
> Importantly, CDNNs are not merely a reparameterisation of VNNs with a temperature parameter. The density matrix transformation introduces inherently new capabilities:
>
> Spectral inversion of discriminability (Theorem 1): For positive β, high-variance components (which are non-discriminable in VNNs) become discriminable. This is not achievable by any scalar rescaling of the covariance matrix.
>
> Multi-scale filter banks: By combining positive and negative β values, CDNNs can discriminate signal differences in both high- and low-variance subspaces simultaneously,  something impossible with VNNs.
>
> Inherent regularisation of singular covariances: The density matrix is always full-rank and positive definite (Proposition 6), even when the covariance matrix is rank-deficient. This is not achievable by trace-normalisation.
>
> Information-theoretic tools: The multi-scale Von Neumann Entropy for covariance matrices (CVNE) is a genuinely new quantity with proven sub-additivity (Theorem 5) that can distinguish singular covariance matrices where classical entropy diverges.
>
> 2. "Lack of clarity and accessibility":
>
> We agree the original text was too compressed. As described in our response to Reviewers Z2dP and HrxG, we have added:
>
> A preamble to Section 2 explaining GSP concepts in plain language and an explicit description of when and why one would use this operator
>
> 3. "Triviality of theoretical analysis":
> We respectfully disagree. While permutation equivariance (Theorem 2) is indeed a standard verification, the remaining theoretical contributions are novel:
>
> Theorem 1 (Composite Lipschitz Conditions): This is not a routine adaptation. We derive the Lipschitz constant through the composition of the density transformation with the spectral filter, revealing the crucial insight that β flips the discriminability structure , mapping high-variance (non-discriminable) covariance eigenvalues to low-density eigenvalues where the filter can discriminate them. This spectral inversion has no analogue in standard GNN stability theory.
>
> Theorem 3 (Network Stability Bound): The bound contains explicit penalty terms F(β, C, δC) and exp(1_{β<0}|β|‖C‖) that quantify the cost of negative β , showing that positive β is strictly more stable. This is derived via Duhamel's formula for matrix exponential perturbations (Lemma 1), which requires a bespoke analysis specific to our density transformation and does not follow from existing GNN perturbation results.
>
> Theorem 5 (Sub-additivity of CVNE): The proof that our covariance entropy satisfies sub-additivity requires a non-trivial regularisation trick (shifting eigenvalues so the partition function is ≥ 1) that is specific to the covariance density setting.
>
> These results collectively establish that CDNNs have a principled, tuneable stability-discriminability trade-off that is absent in VNNs and not derivable from existing GNN theory.
>
> 4. "Empirical results difficult to interpret":
>
> We have added explicit interpretation paragraphs after each results table, linking empirical gains to theoretical properties. For example:
>
> “We attribute the improved performance of CDNN over a standard VNN to the versatility of the multi-scale filter bank, capturing information at different scales”
>
> “The large gains on BCI-2A 4-class (50.4% vs 30.1% for VNN) are explained by the multi-scale filter bank's ability to capture patterns at different spectral scales of the cross-subject covariance matrix (Figure 3).”

---

### Review · Reviewer_Z2dP · 2026-02-01

**Summary Of Contributions:**

**Summary**
This paper introduces CDNNs. They use sample covariance matrix of the data as a central operator and transform the covariance matrix into a density matrix and use it to define filtering operations. A parameter controls how much the model trades stability for discriminability. The method is evaluated on financial forecasting and EEG classification tasks. The results show improvements over covariance neural networks and several graph-based baselines.

Overall, the paper is technically strong and the experimental results are promising. However, I found the presentation difficult to follow without a strong background in graph signal processing.

**Strengths**

- Novel and original idea by combining covariance-based models with density-matrix constructions.

- Deep theoretical analysis that shows care in studying stability and robustness.

- Experimental results on finance and EEG tasks are strong and suggest practical relevance.

- The method improves over prior covariance neural networks and shows good robustness to noise.

**Weaknesses and Suggestions**

- Introduction and Background (Graph Signal Processing). The paper heavily relies on concepts from graph signal processing (GSP), such as graph shift operators, spectral filtering, and transferability. However, the current introduction assumes substantial prior knowledge. For readers not specialized in GSP, it is hard to build intuition. I strongly recommend adding a more pedagogical introduction or an intuitive subsection that explains the main ideas of GSP and covariance-based filtering in simpler terms, possibly with a small running example.

- In the experimental section, it is not always clear: what exactly is the input to each model, its dimensionality,  what is the output of the CDNN layers, how temporal data is transformed before classification. Also, how the covariance matrix is computed and reused at training and test time. A simple diagram/example could be used to improve readability.

- Computational Complexity and Practical Cost While runtime is reported, I miss a clearer discussion of: computational complexity with respect to number of nodes / channels, memory cost of the density matrix and matrix exponential, how these costs compare to non-graph baselines.

- Baselines Beyond Graph-Based Models Most comparisons are against graph-based or covariance-based models.  I recommend including additional strong non-graph baselines, such as: TABPFN or similar tabular foundation models (where applicable),  transformer-style baselines with comparable parameter counts, other non-graph time-series models.

**Audience:**

Yes

**Audience Explanation:**

Yes. Researchers familiar with graph signal processing and machine learning for graphs would likely be interested in this work.

**Broader Impact Concerns:**

Nothing at this point

**Claims And Evidence:**

Yes

**Claims Explanation:**

The main claims of the paper are supported by theoretical analysis and experimental results

**Requested Changes:**

**Weaknesses and Suggestions**

- Introduction and Background (Graph Signal Processing). The paper heavily relies on concepts from graph signal processing (GSP), such as graph shift operators, spectral filtering, and transferability. However, the current introduction assumes substantial prior knowledge. For readers not specialized in GSP, it is hard to build intuition. I strongly recommend adding a more pedagogical introduction or an intuitive subsection that explains the main ideas of GSP and covariance-based filtering in simpler terms, possibly with a small running example.

- In the experimental section, it is not always clear: what exactly is the input to each model, its dimensionality,  what is the output of the CDNN layers, how temporal data is transformed before classification. Also, how the covariance matrix is computed and reused at training and test time. A simple diagram/example could be used to improve readability.

- Computational Complexity and Practical Cost While runtime is reported, I miss a clearer discussion of: computational complexity with respect to number of nodes / channels, memory cost of the density matrix and matrix exponential, how these costs compare to non-graph baselines.

- Baselines Beyond Graph-Based Models Most comparisons are against graph-based or covariance-based models.  I recommend including additional strong non-graph baselines, such as: TABPFN or similar tabular foundation models (where applicable),  transformer-style baselines with comparable parameter counts, other non-graph time-series models.

---

> ### Author Response · Authors · 2026-02-14
>
> We thank Reviewer Z2dP for the thorough and constructive review and for recognising the novelty of our contribution. We address each concern below.
>
> 1. Introduction to GSP:
>
> We have now changed the introduction to lead with the main problem and introduce important concepts from GSP.
>
> We have also expanded Section 2 with a new preamble that explains the core ideas of graph signal processing and covariance-based filtering in accessible terms before presenting formal definitions. This includes:
>
> A plain-language explanation of what a graph shift operator does (propagating/mixing signals along graph edges, analogous to a time-shift in classical DSP)
>
> An intuitive explanation of why the covariance matrix is a natural GSO when no graph is available
>
> A brief explanation showing how filtering with the covariance matrix versus the density matrix differs in which signal components are emphasised
>
> We believe this makes the paper substantially more accessible to readers outside the GSP community.
>
> 2. Computational complexity discussion:
>
> We have added a new subsection (4.3)  summarising the scalability analysis (previously only in Appendix A.5):
>
> The matrix exponential costs O(m^3) but is computed once and cached, so per-step cost is identical to VNN (O(Km^2) per filter of order K).
> Memory cost is O(m^2) for the density matrix, identical to VNN.
> With caching, CDNN runtime matches VNN even for m > 1000 channels (see Appendix A.5, Figure 7).
> Compared to non-graph baselines: EEGNet's per-epoch cost is 4× higher than CDNN (Table 2).
>
> 3. How the covariance matrix is computed:
>
> We have added a small subsection (4.1) explaining the computation of the covariance matrix and refereeing the reader to the Appendix A.2 for full experimental details.
>
>
> 4. Non-graph baselines:
>
> We note that:
>
> For EEG classification, we already include EEGNet (Lawhern et al., 2018), which is the standard non-graph CNN baseline for BCI and has been shown to outperform models such as tabular and recurrent models in EEG signal analysis. We also include CSP+LDA, a classical non-graph approach.  Further we are using a Graph Attention Network operating on a fully connected graph, this reduces the network to just being a normal transformer (Joshi et al. 2025).
>
> For financial forecasting, again, since we are using a fully connected graph for the GAT model it reduces to just a normal transformer. We note that our Hybrid model (covariance density filter + attention) already demonstrates the benefit of combining CDNN with attention-based architectures.

---

### Decision · Action_Editor_qaB2 · 2026-03-07

**Recommendation:** Accept as is

**Audience:**

Yes

**Audience Explanation:**

The paper will be interesting for  the ML community working with graphs and signal processing.

**Claims And Evidence:**

Yes

**Claims Explanation:**

The authors are unanimous to acknowledge that this paper is interesting, all claims are well supported. The scope and goals of the paper are clear and comparison w.r.t. baselines  are compelling. The presentation has been improved.